# From Sparse Dependence to Sparse Attention: Unveiling How Chain-of-Thought Enhances Transformer Sample Efficiency

**Kaiyue Wen**[*][†]
Stanford University
`kaiyuew@stanford.edu`

**Huaqing Zhang**[*]
IIIS, Tsinghua University
`zhanghq22@mails.tsinghua.edu.cn`

**Hongzhou Lin** [‡]
Amazon
`hongzhou.lin89@gmail.com`

**Jingzhao Zhang**
IIIS, Tsinghua University
Shanghai AI Lab
Shanghai Qizhi Institute
`jingzhaoz@mail.tsinghua.edu.cn`

## Abstract

Chain-of-thought (CoT) significantly enhances the reasoning performance of large language models (LLM). While current theoretical studies often attribute this improvement to increased expressiveness and computational capacity, we argue that expressiveness is not the primary limitation in the LLM regime, as current large models will fail on simple tasks. Using a parity-learning setup, we demonstrate that CoT can substantially improve sample efficiency even when the representation power is sufficient. Specifically, with CoT, a transformer can learn the function within polynomial samples, whereas without CoT, the required sample size is exponential. Additionally, we show that CoT simplifies the learning process by introducing sparse sequential dependencies among input tokens, and leads to a sparse and interpretable attention. We validate our theoretical analysis with both synthetic and real-world experiments, confirming that sparsity in attention layers is a key factor of the improvement induced by CoT. [1]

## 1 Introduction

Chain-of-thought (CoT) has proven to be a powerful technique for enhancing reasoning in large language models Wei et al. (2022); Kojima et al. (2022). By instructing the model to break complex problems into smaller, manageable steps, CoT facilitates more efficient reasoning and better generalization, particularly in algorithmic and logical tasks Nye et al. (2022); Lewkowycz et al. (2022); Wang et al. (2023b). Building on this, performance can be further improved through multi-step prompting and multi-path sampling techniques Chowdhery et al. (2023); Wang et al. (2022b); Zhou et al. (2023a); Zhang et al. (2023); Fu et al. (2023).

This focus on CoT within in-context learning has since expanded to more structured learning approaches Yao et al. (2024); Besta et al. (2024). By adding reasoning examples of CoT style to the instruction-tuning dataset, models enhance their problem-solving abilities more effectively than relying solely on CoT during prompting Zelikman et al. (2022); Chung et al. (2024). As a result, CoT is now shaping a new paradigm in language model development, marking a shift from simply scaling data Kaplan et al. (2020); Hoffmann et al. (2022) to focusing on advanced reasoning strategies Lightman et al. (2024), which leads to more effective learning outcomes.

---

[*]These authors contributed equally.

[†]A large part of this work was done while Kaiyue was at Tsinghua University.

[‡]This work is independent of and outside of the work at Amazon.

[1]Our code is available at `https://github.com/zhqwqwq/Learning-Parity-with-CoT`.

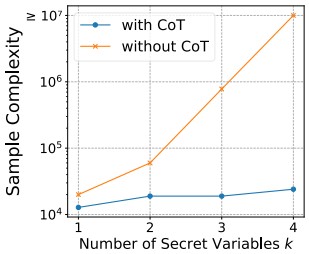 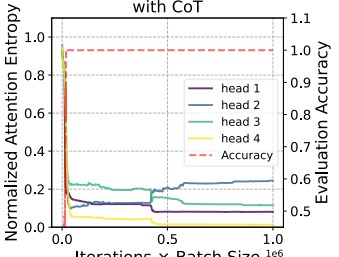 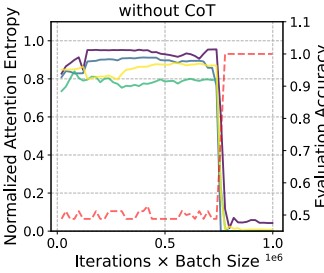

(a) Sample complexity of 4-layer 4-head transformer on parity with $n = 30$.

(b) Normalized attention entropy and evaluation accuracy curves of 4-layer 4-head transformer on $(n = 30, k = 3)$ parity problem.

Figure 1: (a) We show that, without Chain-of-Thought (CoT), the sample complexity for training transformers to learn the parity function grows exponentially with the hardness parameter $k$. In contrast, utilizing CoT significantly improves sample efficiency. (b) We also show that the sparsity of attention layers, measured by normalized entropy (1), is crucial in the parity learning experiment. In both CoT and non-CoT scenarios, as the attention layers become sparser—indicated by a rapid decrease in normalized entropy—a corresponding jump in evaluation accuracy occurs.

While CoT's success is well-established, understanding why it works is still a hotly debated topic Saparov & He (2023); Prystawski et al. (2024). Recent theoretical studies suggest that CoT enhances a model's expressiveness, increasing its representational capacity when the sequence is long enough Feng et al. (2023); Li et al. (2024c). However, expressivity alone does not guarantee success. Large language models often struggle with simple tasks—like counting the number of 'r's in "strawberry"—when not using CoT. Given the increasing model sizes, it seems unlikely that such tasks are inherently inexpressible. This discrepancy calls for a deeper study of generalization, hinting the true power of CoT may lie beyond the expressiveness.

In this paper, we study the benefit of CoT from the sample efficiency perspective. We provide concrete examples where expressiveness is not the limiting factor; that is, the function can be expressed both with and without CoT by the transformers. We demonstrate, both in theory and in practice, that without CoT, the learning requires exponentially more samples comparing to with CoT. Further, we show that CoT sequences introduce sparse sequential dependence, thereby enhancing the sparsity in the attention layers. We then show that transformers can efficiently optimize and generalize on such sequences. We summarize our contributions as follows.

1. Theoretically, we show that while the parity problem can be expressed by a 1-layer transformer, learning requires exponentially many samples without CoT when the number of parameters is limited (Theorem 1 and 2). Meanwhile, for CoT data with sparse sequential dependence, a 1-layer transformer can learn the parity function and faithfully represent the sparse dependence in its attention pattern with almost linear samples with respect to sequence length(Theorem 3).

2. Empirically, we verify our analysis that training on parity function with CoT data requires only polynomial samples and will induce sparse and interpretable attention (Figure 3). We further show that evaluating and training on CoT data will also induce sparser attention on real-world dataset GSM-8k on pretrained language models.

## 2 MOTIVATING EXAMPLES: LEARNING PARITY FUNCTIONS

We start with empirically exploring the sample efficiency of training transformers to learn the class of parity functions (Blum et al., 2003) with and without CoT. The parity functions exhibits simple structure but yet hard to learn by traditional networks Abbe & Sandon (2018); Malach & Shalev-Shwartz (2022). Specifically, given a set of $n$ binary variables $b_1, b_2, \ldots, b_n$, a parity function takes $k$ secret variables $b_{i_1}, \ldots b_{i_k}$ and outputs 1 if the sum of these $k$ variables is odd, and 0 if it is even:

$$f(b_1, ..., b_n) = b_{i_1} \oplus b_{i_2} \oplus \cdots \oplus b_{i_k},$$

where $b_i \in \{0, 1\}$ and $\oplus$ is the XOR operator. For example, $f(b_1, b_2, b_3, b_4, b_5) = b_1 \oplus b_2 \oplus b_4$ is a 5-variable parity function with $k = 3$. The function $f$ returns 1 if the sum of $b_1$, $b_2$ and $b_4$ is odd, and 0 otherwise, independently of the value of $b_3$ and $b_5$. Intuitively, the parameter $k$ controls the hardness of the problem. When $k$ increases, the number of possible subsets grows exponentially, making the identification of the correct secret set more challenging.

Given a $n$-variable parity function $f$, we generate sequences of length $n + 2$ as auto-regressive manner: $b_1, \cdots b_n, b_{n+1} := 0$ and $b_{n+2} = f(b_{[1:n]})$. To incorporate CoT, we break the sum of XOR down into $k$ steps and add all the intermediate steps into the sequence, i.e. including $b_{i_1}, b_{i_1} \oplus b_{i_2}$, $\cdots, b_{i_1} \oplus b_{i_2} \oplus \cdots \oplus b_{i_k}$. As a result, the CoT data has length $n + k + 1$. With the example function $f(b_1, b_2, b_3, b_4, b_5) = b_1 \oplus b_2 \oplus b_4$, one sampled sequence would be

$$\text{No CoT} \quad \underbrace{0,1,0,1,0,}_{\text{input}} \underbrace{0}_{\text{[EOS]}}, \underbrace{0}_{\text{answer}} \in \{0,1\}^7, \text{ as } b_1 \oplus b_2 \oplus b_4 = 0 \oplus 1 \oplus 1 = 0.$$

$$\text{With CoT} \quad \underbrace{0,1,0,1,0,}_{\text{input}} \underbrace{0}_{\text{[EOS]}} \underbrace{0,1}_{\text{CoT}}, \underbrace{0}_{\text{answer}} \in \{0,1\}^9, \text{as } b_1 = 0, b_1 \oplus b_2 = 1.$$

Next, we train transformer networks with and without CoT respectively, with a common held-out test set, and compare their sample complexities. The sample complexity is defined as the amount of data a model sees at the first time it achieves perfect validation accuracy. More precisely, we halt the training process when the model reaches 100% validation accuracy and record the total number of training examples used up to this point as the empirical sample complexity.

In Figure 1, we compare the empirical results of learning parity function with and without CoT. The results show that the sample complexity without CoT (orange curve) grows exponentially, while with CoT (blue curve), it increases linearly. In other words, incorporating CoT allows us to learn the task with exponentially fewer samples. Additionally, it suggests that sparsity in the attention layers plays a crucial role in the learning process. To explore why CoT enables transformers to learn more efficiently, we formally define our problem setup below.

## 3 THEORETICAL ANALYSIS

In this section, we provide a formal analysis of the training dynamics of Transformers on the parity problem, both with and without Chain of Thought (CoT). We select the parity problem as our testbed because identifying a set of key variables amidst various confounding ones is a fundamental aspect of many reasoning tasks, and parity serves as an abstraction of this process.

### 3.1 NOTATIONS AND DEFINITIONS

**Parity Problem.** Each token in our setting is either $0$ or $1$. We represent a sequence of length $T$ as $\mathbf{b}[1], \ldots, \mathbf{b}[T]$. A parity function is a function of form $\text{parity}_S(\mathbf{b}) = \oplus_{j \in \mathcal{S}} \mathbf{b}[j]$, where $\mathcal{S}$ is the set of secret indices that is fixed during training and testing, and $\oplus$ is the XOR operator on binary variables. The cardinal of the secret set $k = |S|$ controls the hardness of the problem.

**Definition 1** (Parity Problem $(n, k)$ without CoT). *Given a secret set $\mathcal{S}$ of cardinal $k$, we define the parity problem without CoT as the following distribution of sequences $p_{\text{NoCoT}}$:*

$$\mathbf{b}[i] \sim U(\{0, 1\}), \forall i \in [1, n], \mathbf{b}[n + 1] = 0, \mathbf{b}[n + 2] = \text{parity}_S(\mathbf{b}).$$

*where $\mathbf{b}[1], ..., \mathbf{b}[n]$ are uniformly sampled from $0$ and $1$.*

**Definition 2** (Parity Problem $(n, k)$ with CoT). *Given a secret set $\mathcal{S}$ of cardinal $k$, we define the parity problem with CoT as the following distribution of augmented sequences $p_{\text{CoT}}$:*

$$\mathbf{b}[i] \sim U(\{0, 1\}), \forall i \in [1, n], \mathbf{b}[n + 1] = 0, \mathbf{b}[i + 1] = \mathbf{b}[i] \oplus \mathbf{b}[S[i]], \forall i \in \{n + 1, \ldots, n + k\}.$$

*where $S$ is any permutation of $\mathcal{S}$, i.e. $\{S[i] \mid i = n + 1, \ldots, n + k\} = \mathcal{S}$.*

When CoT is provided, the data includes a step-by-step computation of the desired parity function, adding one variable at a time. Note that CoT is not unique for a given secret set since we can arbitrarily permute $\mathcal{S}$, given the commutative property of the XOR operator.

**Transformer Architecture.** To conduct the theoretical analysis, we simplify the Transformer architecture similar to prior works (see e.g. Wang et al. (2024b); Nichani et al. (2024); Li et al. (2023b)). More precisely, we drop all the layer norms or batch normalization; simplify the positional embedding; use a square matrix to represent the attention layer and concatenate the residual branch in a Densenet fashion Huang et al. (2017). After simplification, the network still has strong expressive power like the standard Transformers.

In a standard Transformer, an embedding layer matches each vocabulary token into a dense vector of size $d$ and then adds a positional embedding. In our case, we only have boolean tokens and the sequence has constant length $T$. Hence, we simply freeze $2T$ unit vectors as the embedding vectors:

**Definition 3** (Embedding Module). *For any position $i \in [T]$, we sample two embedding vectors $e_{i,0}$ and $e_{i,1}$ uniformly at random from unit hypercube $\mathrm{U}\left(\{\frac{1}{\sqrt{d}}, -\frac{1}{\sqrt{d}}\}^d\right)$. These embedding vectors are frozen during training. Then for any binary sequence $\boldsymbol{b} \in \{0,1\}^T$, the embedding is defined as*

$$\mathbf{E}\left(\boldsymbol{b}\right) = [e_{(i,\boldsymbol{b}[i])}]_{i\in[T]} \in \mathbb{R}^{d\times T}.$$

*Due to the properties of the hypercube, with high probability, the embedding vectors are near-orthogonal with each other.*

Next, we define the attention layer. In standard architecture, the attention layer is derived from the product of query and key matrices, i.e. $QK^T = \mathrm{X}^T W^Q (W^K)^T \mathrm{X}$. In our case, we directly use a full matrix $\mathrm{X}^T A \mathrm{X}$ to parameterize it, which has the same representation power.

**Definition 4** (Attention Module). *Given an input matrix $\mathrm{X} \in \mathbb{R}^{d\times T}$ and attention weight $A$, we define the attention module as*

$$\mathcal{A}(\mathrm{X}) = \mathrm{X}\,\mathrm{softmax}(C + \mathrm{X}^T A \mathrm{X}),$$

*where $C \in \mathbb{R}^{T\times T}$ is the autoregressive mask with value $-\infty$ on the lower triangular matrix.*

The attention module is then followed by a fully connected layer with ReLU activation:

**Definition 5.** *We define the FFN function with width $2m$ over a matrix $\mathrm{X} \in \mathbb{R}^{2d\times T}$ as*

$$\mathrm{FFN}(\mathrm{X}) = h^T \mathrm{ReLU}(W X).$$

*Here $W \in \mathbb{R}^{2m\times 2d}, h \in \mathbb{R}^{2m\times o}$ with $o$ being the output dimension and $\mathrm{ReLU}$ is element-wise.*

Finally, we concatenate the modules into a simplified Transformer block.

**Definition 6** (Simplified Transformer Block). *We define a simplified Transformer block as,*

$$\mathcal{T}(\boldsymbol{b}) = \mathrm{FFN}\left(\begin{bmatrix} \mathbf{E}\left(\boldsymbol{b}\right) \\ \mathcal{A}\left(\mathbf{E}\left(\boldsymbol{b}\right)\right) \end{bmatrix}\right),$$

*where $\mathbf{E}$ is the embedding module, $\mathcal{A}$ is the attention module and $\mathrm{FFN}$ the fully connected layer. An $L$-layer Transformer consists of a composition of $L$ such blocks, with the embedding module $\mathbf{E}$ appearing only in the initial layer. The intermediate dimension is set to $d$, while the output dimension of the final layer is 1.*

Compared with the standard residual structure, we use a Densenet structure to concatenate the residual branch with identity branch. Again, this transformation does not affect the representation power of the network, which is the standard practice in previous theoretical analysis (see e.g. Wang et al. (2024b); Nichani et al. (2024)).

**Loss function.** To simplify our analysis, we use hinge loss $\ell(\hat{y}, y) = \max\{(-1)^y \hat{y} + 1, 0\}$ as the loss function. We define the next token prediction loss of a boolean sequence $\boldsymbol{b}$ as $L(w) = \ell(\mathcal{T}^{(L)}(\mathbf{b})[n+1], \mathbf{b}[n+2])$ and $L(w) = \sum_{i=n+1}^{n+k} \ell(\mathcal{T}^{(L)}(\mathbf{b})[i], \mathbf{b}[i+1])$ for with and without CoT setup respectively, where $w$ denotes all the trainable parameters.

### 3.2 Exponential Sample Complexity without CoT

We now present analysis in the no CoT setup. First, we show that the parity problem is easy to represent with the simplified Transformer architecture:

**Theorem 1** (Easy to Represent). *Consider the Transformer model defined in Definition 6, for any $\delta < 0.1$ and large enough $n$, when the number of secret indices $k$ is in $[n/\log^5(n/\delta), n/\log^4(n/\delta)]$, with probability at least $1 - \delta$ over the randomness of embedding $e$, there exists a weight configuration of the Transformer with dimension $d = \Theta(k\log(n/\delta))$ and width $2m = O(k)$ with $\Theta(\log n)$ precision of the weights and activations, such that $d^2 + dm = o(nk/\log n)$, and it achieves perfect accuracy on the parity problem $(n, k)$ without CoT, i.e.*

$$\forall \boldsymbol{b} \sim p_{\mathrm{NoCoT}}, \mathrm{sgn}\left(\mathcal{T}(\boldsymbol{b})[n+1]\right) = (-1)^{\boldsymbol{b}[n+2]+1}.$$

In other words, the model possesses sufficient expressive power to represent any parity function, even in the absence of the chain of thought. Therefore, we are in the *representational-sufficient* regime where expressiveness is not the bottleneck. The proof of this statement is based on random matrix theory and concentration inequalities, which we defer to Appendix B.2.

While the function is expressible, this does not guarantee that the solution can be easily found. In fact, we show that achieving a perfectly accurate solution using a gradient-based optimization method requires an exponential number of samples in $k$ assuming the memory, i.e. space complexity, to perform optimization is bounded throughout the training.

**Theorem 2** (Hard to Learn). *For any randomly initialized simplified Transformer model with constant layers, when the embedding dimension $d$ and width $2m$ satisfies $d^2 + dm = o(nk/\log(n))$, for any constant number of passes $q$, when the model is trained with $q-$pass stochastic gradient descent with $O((d^2 + dm)\log n)$ memory, the sample complexity required to learn the parity problem $(n, k)$ without CoT to any nontrivial accuracy $a > 50\%$ with nontrivial probability $p > 50\%$ is $2^{\Omega(k)}$.*

The proof is deferred to Appendix B.3. We use results from the classical online learning communities (Lyu et al., 2023) to show that in the regime where the memory required to perform the training is less than $nk$, the model can't effectively store information about enough samples inside the parameters during training, and hence can't infer the secret indices effectively. Here the parameters $k$ denotes the size of the secret set $\mathcal{S}$ and the sample complexity grows exponentially when $k$ grows.

### 3.3 POLYNOMIAL SAMPLE COMPLEXITY WITH COT

The result in Theorem 2 presents a seemingly negative outcome for learning algorithmic reasoning, as the model requires exponentially many samples relative to $k$. However, in our main result, Theorem 3, we will demonstrate that the sample complexity can be significantly reduced if a step-by-step derivation is provided. In other words, CoT is much more sample efficient, where model can effectively learn complex relationships as long as each token depends on only a few previous tokens.

We will use the following initialization of the 1-layer Transformer model. The attention is initialized to be uniform ($A = 0$) and the contribution of attention output is initialized to be zero ($W_{r,d+1:2d} = 0$). The parameter $h$ in the FFN and the word embedding $\mathbf{E}$ is fixed during training.

**Assumption 1** (Initialization). *At initialization,*

$$\forall r \in [2m], A = 0, W_{r,d+1:2d} = 0, W_{r,1:d} = \sum_{i=n+1}^{n+k} \sum_{b=0}^{1} \nu_{r,i,b} e_{i,b}, h_{1:m} = \frac{1}{2m}, h_{m+1:2m} = -\frac{1}{2m}.$$

*Here $\nu_{r,i,b}$ is independent random variable sampled uniformly from $\{-\epsilon, \epsilon\}$.*

We can then train the model with stochastic gradient descent (SGD) and get the following theorem.

**Theorem 3** (Easy to Learn with CoT). *For any constant $\delta \in (0,1)$, when the number of secret indices $k$ is in $[n/\log^5(n/\delta), n/\log^4(n/\delta)]$, with probability $1 - \delta$, a randomly initialized simplified Transformer (see Assumption 1) with $o(nk)$ parameters trained for constant steps using mini-batch SGD with $\tilde{O}(n)$ samples using appropriate hyperparameters (see Assumption 2) can reach perfect accuracy on a parity problem $(n, k)$ with CoT,*

$$\forall \boldsymbol{b} \sim p_{\text{CoT}}, i \in \{n+1, \ldots, n+k\}, \operatorname{sgn}(\mathcal{T}(\boldsymbol{b})[i]) = \boldsymbol{b}[i+1].$$

*Furthermore, after training, the attention pattern is interpretable and one-hot in the sense that, for any $\boldsymbol{b} \sim p_{\text{CoT}}, i \in \{n+1, \cdots, n+k\}, j \leq i$,*

$$\left|\operatorname{softmax}(C + \mathbf{E}(\boldsymbol{b})A\mathbf{E}(\boldsymbol{b}))[j,i] - \mathbf{1}(j = S[i])\right| < 1/n^8.$$

*Moreover, the result still holds even if all the weights and activations are in $\Theta(\log n)$ precision.*

This theorem indicates that the attention module can successfully extract the sparse sequential dependencies in the CoT data and faithfully represent it in the attention pattern, which is validated in our experiments (see Figure 3). It is also amongst the first optimization dynamics analysis of Transformers using finite-sample gradients rather than population gradients. We delay the full proof of the theorem to Appendix B.5.

*Proof Sketch.* In our analysis, the dynamics of the model includes three key phases. In the first phase, the weight of the FFN layers become correlated with the embeddings associated to the secret indices. In the second phase, this correlation caused the attention module to receive a strong signal to amplify the attention weight on the corresponding secret indices at each position. The attention pattern will become one-hot after this step. Finally, in the third phase, the FFN layers learn the correct mapping to the output, utilizing both the embedding at the current token and the retrieved embedding from the secret indices as indicated by the attention.

**Phase 1. Configuring the FFN** At initialization, as the FFN weight corresponds to the attention output ($W_{r,d+1:2d}$) is initialized to be zero. The set of neurons activated at each position $i \in [n + 1, n+k]$ is solely determined by the word embedding $e_{(i,\mathbf{b}[i])}$ at the position. Because $e_{i,b}$ are nearly orthogonal, it holds that $\langle e_{i,\mathbf{b}[i]}, W_{r,1:d} \rangle \approx \nu_{r,i,b}$ for all $i$ and $r$ with high probability. As a result, for a fixed $i$ and $\mathbf{b}[i]$, the set of activated neurons at position $i$ is $\{r \mid \nu_{r,i,\mathbf{b}[i]} > 0\}$. Therefore, at initialization, the MLP can be viewed as an ensemble of multiple linear functions specialized to each position and boolean value. We can show that when the learning rate is small, the set of activated neurons at each position remains the same through the training process. Hence, we can conceptually view the FFN function as a set of different linear functions applied independently at each position and value. We will denote this set of linear weight as $\kappa_{t,i,b}$ (formally defined in Lemma 18).

As the attention weight matrix $A$ is initialized as zero, the attention pattern will be uniform at initialization. Notice that in the parity data with CoT, the only position whose value correlates with $\mathbf{b}[i + 1]$ when conditioned on $\mathbf{b}[i]$ is $S[i]$. This suggests that the linear weight $\kappa_{1,i,\mathbf{b}[i]}$ will have a stronger correlation with the embedding $e_{(S[i],\mathbf{b}[j])}$ than other embeddings.

This step crucially relies on the sparse dependency in the CoT data. The strong linear correlation will not be present in the data without CoT when $k > 1$, as every token $\mathbf{b}[i]$ for $i \in [n + 1]$ will be uncorrelated with the desired output $\mathbf{b}[n + 2]$ in such case.

**Phase 2. Learning the Sparse Attention** At the second step, as the FFN weight corresponds to the attention output is no longer zero, the attention weight matrix $A$ will receive a non-zero gradient. By the chain rule, the gradient corresponding to how the attention from $i-$th token attends to the $j-$th token $\mathrm{softmax}(C + \mathrm{X}^T A \mathrm{X})[j, i]$ will be larger when the approximate contribution of the embedding of the $j-$th token to the output at the $i-$th token $\langle \kappa_{1,i,\mathbf{b}[i]}, e_{(j,\mathbf{b}[j])} \rangle$ is larger. This suggests that the attention will be amplified on the index that has a strong correlation with the output at the current token, which is the secret index $S[i]$. This step will make the attention pattern approximately one-hot as in the theorem statement.

**Phase 3. Learning the Output Mapping** At the final step, as the attention pattern is one-hot, the FFN layer only needs to learn a mapping from the embedding at the current token and the embedding at the secret index to the output. As the mapping is linear conditioned on the embedding at the current token, the FFN layer can learn the correct mapping within a single step. This step will make the model reach perfect accuracy on the parity problem with CoT.

## 4 EMPIRICAL EXPERIMENTS

In this section, we validate and extend our theoretical findings through comprehensive experiments in the following three aspects: First, we empirically confirm that CoT reduces the sample complexity of transformers in learning the parity problem. Second, we conduct multi-pass training of transformers without CoT, as a complement to the established lower bound (Theorem 2) which only applies to constant-pass training. The results indicate that multi-pass training indeed improves the models' ability to learn parity problems. Third, we conduct experiments on the GSM8K dataset (Cobbe et al., 2021) to show that CoT introduces sparse sequential dependence on real-world training data. Although necessary simplifications on transformer are made in Section 3 to develop theoretical results, we use the standard transformer architecture in the subsequent experiments following the GPT-2 architecture (Radford et al., 2019) with trainable position embeddings unless otherwise specified.

### 4.1 PARITY LEARNING WITH MULTI-LAYER TRANSFORMERS

In this section, we conduct an ablation study on the sample complexity of transformers with standard GPT-2 architectures and one-pass training, with and without CoT.

**Ablation study on sample complexity (Figure 2).** We train transformers on parity problem with $n = 30$ and $k = 1, 2, 3, 4$, with and without CoT, varying layers and heads from 1 to 4. We choose

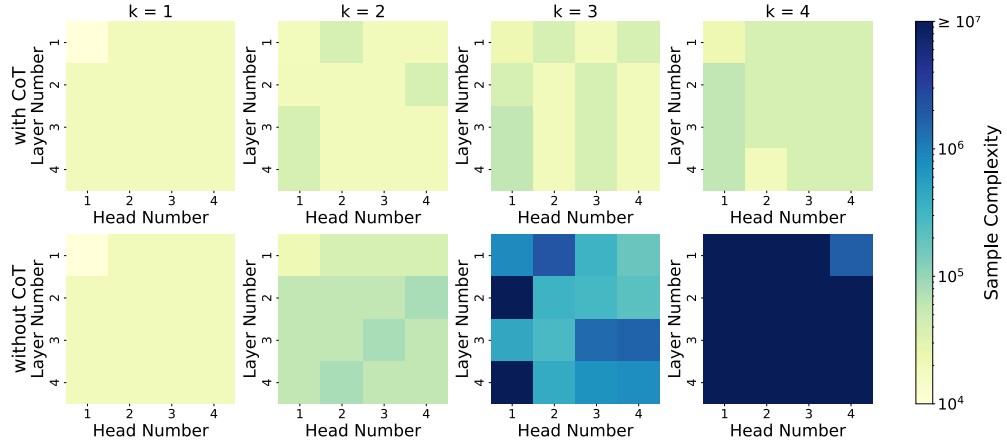

(a) Sample complexity of transformers with varying numbers of layers and heads on parity problem with n=30

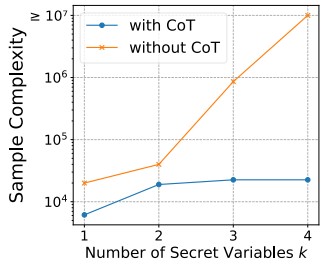

(b) Sample complexity of 1-layer 1-head transformer on parity problem with $n = 30$

(c) Sample complexity of 4-layer 4-head transformer on parity problem with $n = 30$

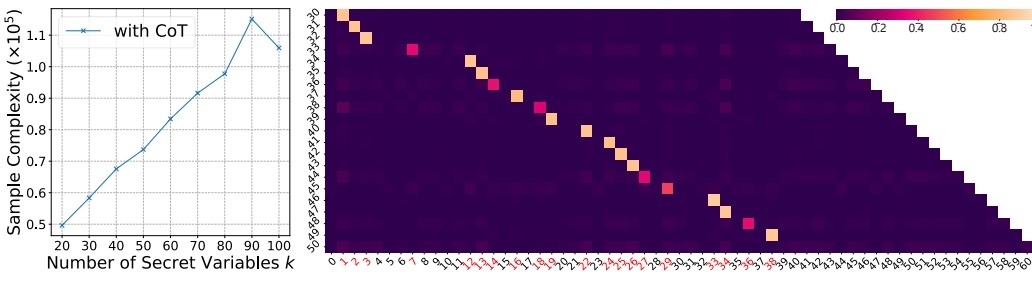

(a) Sample complexity of 1-layer 1-head transformer on parity problem with $n = 100$ using CoT.

(b) The attention pattern of 1-layer 1-head transformer trained on $(n = 40, k = 20)$ parity problem, detailed in Section 4.1.

Figure 3: (a) When $n$ is fixed, the sample complexity of learning parity with CoT grows approximately linearly with $k$. (b) The attention pattern learned by the transformer with CoT is **interpretable**, as the $i$-th output token of CoT predominantly attends to secret index $S[i]$.

the best performer across learning rates from $6 \times 10^{-5}, 8 \times 10^{-5}$, and $1 \times 10^{-4}$. At each step, a fresh batch of training data is sampled, with a maximum budget of $10^7$ samples. We record the number of samples seen by the model before reaching an evaluation accuracy of 1 as the sample complexity.

**Results.** Figure 2 shows that training with CoT [first row in Figure 2 (a)] consistently achieves better sample efficiency than training without CoT [second row in Figure 2 (a)]. Moreover, for a fixed configuration of heads and layers, the sample complexity without CoT grows exponentially with the parameter $k$. In contrast, CoT greatly reduces the sample complexity, showing an exponential improvement across different numbers of heads and layers. These findings are consistent with our theoretical analysis.

**Training with CoT induce sparse and interpretable attentions (Figure 3).** We evaluate the sample complexity of a 1-layer, 1-head transformer on larger parity problems ($n = 100$ and $20 \leq k \leq 100$). Without CoT, training exceeds the sample budget when $k \geq 4$. However, with CoT, we can successfully train for any $k$. Figure 3b shows the attention pattern of a 1-layer, 1-head transformer trained on a parity problem with $(n = 40, k = 20)$ using CoT on a random query. In

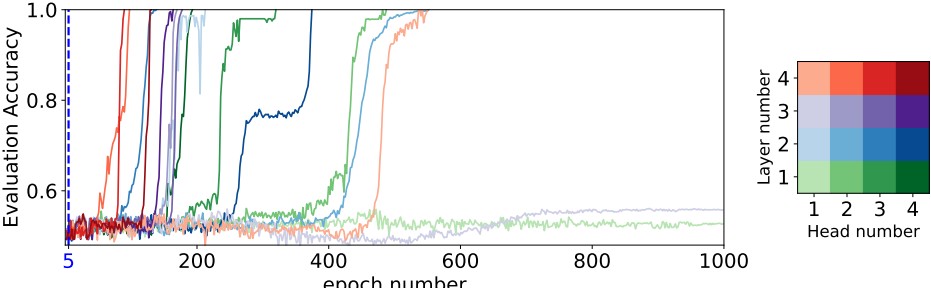

Figure 4: Evaluation accuracy of transformers on the $(n = 20, k = 6)$ parity problem **without** CoT. The model is trained on a dataset of $10,000$ samples for $1,000$ epochs. Almost all layer-head configuration achieve perfect evaluation accuracy. Adding more heads is more effective than adding layers. The blue dashed line marks the **with** CoT setup, which achieves perfect accuracy in 5 epochs.

this case, CoT processes the secret variables in ascending order. The $i$-th row illustrates the attention weight of the $i$-th token attending to the previous ones, with indices corresponding to secret variables highlighted in red.

**Results.** In Figure 3a, we show that training with CoT succeed with large $n$ and $k$. Moreover, the empirical result suggests that the sample complexity of learning parity with CoT grows approximately linearly with $k$. This shows a clear contrast to without CoT setting where the required samples explodes exponentially. To validate our theorem on learning the sparse dependence, we show in Figure 3b that the attention pattern learned by the transformer with CoT is interpretable. Specifically, the $i$-th output token of CoT primarily focuses on the $i$-th secret variable, with minimal attention given to other tokens.

## 4.2 MULTI-PASS TRAINING IMPROVES PARITY LEARNING WITHOUT COT

In our theory, we only establish lower bounds for the sample complexity of learning parity problems with constant pass SGD training. To complement the established lower bounds for transformers learning parity (Theorem 2), we conduct empirical experiments by training transformers without CoT using **multi-pass training**. We make two observations below. First, the results demonstrate that training with repeated data can help Transformers learn the parity function, although this process still consumes significantly more computation (epochs) than trained on CoT data (Figure 4). Second, a key difference between one-pass and multi-pass training is the development of **sparse attention** (see Figure 5 Right), similar to the role of CoT shown in the previous section. This shows that the development of sparse attention is crucial even in when training without CoT on the parity data.

**Experiment protocols.** On $(n = 20, k = 6)$ parity problem, we conduct multi-pass training without CoT. In Figure 4, the models are trained with $10^4$ samples for 1000 epochs, with the number of layers and heads ranging from 1 to 4. In Figure 5, we compare the training of 4-layer 4-head transformer on $5 \cdot 10^4$ and $10^6$ samples respectively. The learning rate is initialized $10^{-4}$.

**Normalized attention entropy.** To measure attention sparsity, we introduce the concept of *normalized attention entropy* for each attention head (illustrated in Figure 5 Right). Let $\mathcal{P}^{\ell,h}(\mathbf{x})[i]$ denote the attention score distribution produced by the $h$-th head in the $\ell$-th layer at the $i$-th token of input $\mathbf{x}$. The normalized attention entropy for the input $\mathbf{x}$ is then defined as:

$$\text{Ent}(\mathbf{x}; \ell, h) = \min_{i \geq 2} \frac{H(\mathcal{P}^{\ell,h}(\mathbf{x})[i])}{\log i} \in [0, 1], \tag{1}$$

where $H(\mathcal{P}) = -\int \log(\mathcal{P})d\mathcal{P}$ is the entropy of distribution $\mathcal{P}$. The normalization term $\log i$ represents the entropy of a uniform distribution over $i$ tokens, account for the varying context length. The minimum is taken over different tokens in $\mathbf{x}$ since attention heads may specialize in extracting information for specific tokens. As a result, a lower normalized attention entropy indicates a sparser attention pattern. To compute the normalized attention entropy for each head, we would average the normalized entropy across all question-answer pairs in the validation set.

**Results.** As shown in Figure 4, when trained on $10^4$ samples, which only accounts for a small portion ($\sim 1\%$) of all possible inputs ($2^{20} \approx 10^6$), most of the transformer architectures we examined achieve perfect evaluation accuracy given sufficient epochs. While the $k = 6$ problem is intractable with one-pass training without CoT, these results demonstrate that multi-pass training can indeed enhance learning in the no-CoT setup. However, CoT is by far the most effective accelerator, achieving perfect accuracy in just 5 epochs, significantly outperforming the multi-pass no-CoT training.

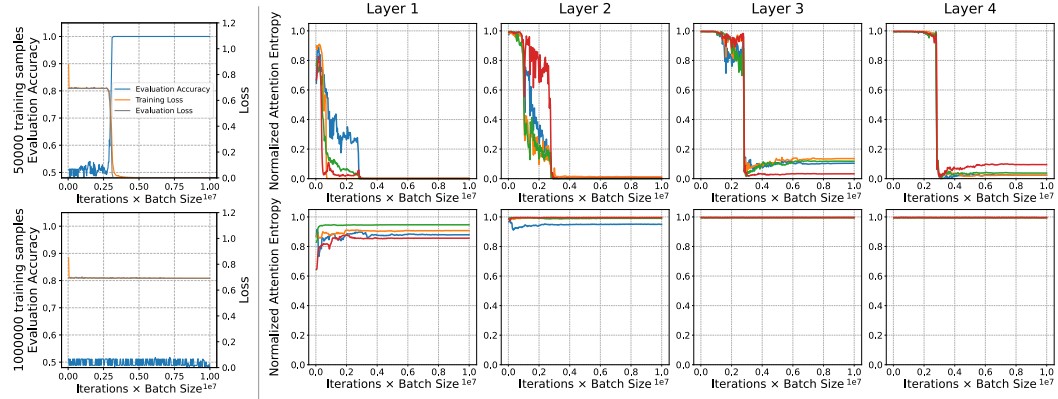

Figure 5: 4-layer 4-head transformer trained on the $(n = 20, k = 6)$ parity problem without CoT using multi-pass training, detailed in Section 4.2. When trained on a small dataset of 50000 samples, the model achieves perfect evaluation accuracy (Top), accompanied by a significant decrease in entropy. Surprisingly, when trained on an even larger training set with 1000000 samples, the model fails to learn (Bottom), and both the training loss and the normalized attention entropy remain elevated. This shows that the development of attention sparsity may improve optimization.

Although learning without CoT is less efficient, it offers a "slow-mode" trajectory. In Figure 5 (Top), the no-CoT loss function initially plateaus before eventually dropping to zero. During this plateau, the entropy in the attention layers continues to decrease, indicating that feature learning is occurring gradually. When all the attention heads become sparse, a transition phase occurs in the loss and accuracy: the loss drops to zero, and evaluation accuracy jumps to 1. In contrast, in the failure case shown in Figure 5 (Bottom), the entropy of the attention layers remains high throughout.

This experiment on no-CoT confirms that sparse attention is crucial for parity learning. As previously demonstrated in Figure 1, CoT accelerates learning by quickly inducing sparsity. This suggests that CoT not only improves the sample efficiency but also improve the optimization landscape by facilitating sparsity, due to the introduction of sparse dependencies added in the intermediate steps.

### 4.3 CoT Induces Sparsity on Real-World Data

Now we move from the synthetic parity problem to real world experiment on GSM8K dataset of grade-school math word problems (Cobbe et al., 2021). We observe that:

1. Real-world CoT data also exhibits sparse sequential dependence, leading to a sparser attention pattern in pre-trained models.

2. Fine-tuning on CoT data further enhances the models' attention sparsity on the input data.

**Experiment protocols.** In Figure 6, we examine two data types: `With CoT`, where inputs from GSM8K dataset are concatenated with ground truth answers that include multiple reasoning steps, and `No CoT`, where inputs are directly concatenated with the final answer. We evaluate two language models: the pre-trained model Qwen2-7B (Yang et al., 2024) and the specialized mathematics model Qwen2-7B-Math (Qwen, 2024) which is fine-tuned from Qwen2-7B on a mathematics-specific corpus with CoT data. We plot the normalized attention entropy (Equation (1)) across different heads. More details can be found in the Appendix Appendix C.1.

**Results.** Unlike the synthetic parity problem, the sequential dependency of real-world data is hard to measure directly. However, it can be inferred from the attention sparsity of pre-trained models when they process such data as input. As shown in Figure 6, comparing the normalized attention entropy of the same pre-trained model Qwen2-7B on different types of data, we can see that the entropy is lower for `With CoT` data compared to `No CoT` data, indicating that real-world CoT data indeed exhibits a sparser structure. Furthermore, on the same `With CoT` data, Qwen2-Math-7B model demonstrates lower attention entropy compared to Qwen2-7B model, suggesting that fine-tuning on CoT data promotes the development of sparser attention patterns in the model.

## 5 Additional Related Work

**Parity Learning.** The most relevant work to ours is Wies et al. (2023), which shows that subtask decomposition enables learning the parity problem with polynomial sample complexity in recurrent neural networks (RNNs). Their learnability results rely on techniques from Wang et al. (2022a), which operate within the NTK linearization regime of RNNs. In contrast, the Transformers an-

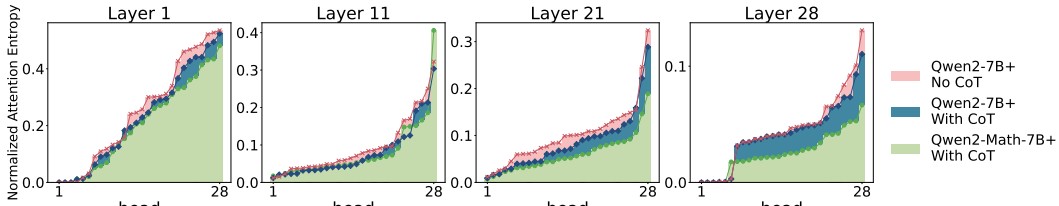

Figure 6: We compare the normalized attention entropy of the pre-trained Qwen2-7B and math-specialized Qwen2-Math-7B models on the GSM8K dataset with and without CoT prompting (Section 4.3). Each bar represents the entropy of an attention head. The Qwen2-7B model exhibits sparser attention when processing CoT data, indicating that real-world CoT data has a sparser structure. The entropy difference between the Qwen2-7B and Qwen2-Math-7B model suggests that fine-tuning on CoT data promotes the development of sparser attention patterns of the model.

alyzed in our work exhibit feature learning and identify the sparse secret set within the attention module. On the positive side, a line of works that studies optimization dynamics of neural networks on parity (Kou et al., 2024; Barak et al., 2023; Edelman et al., 2023; Daniely & Malach, 2020; Abbe et al., 2023; 2024b) show that $n^{\Omega(k)}$ samples is sufficient to learn parity, which is close to the statistical query lower bound (Kearns, 1998). On the negative side, it is well-established that learning parity using gradient descent (GD) requires $\Omega(n^k)$ iterations due to the SQ lower bound (Kearns, 1998). For SGD learning, Shalev-Shwartz et al. (2017); Abbe & Sandon (2020) showed that an exponential number of samples is necessary when SGD is hindered by additional noise or when the number of weights updated in each step is constrained. However, it remains unclear whether standard mini-batch SGD will still need exponential sample complexity. In our paper, we take the alternative approach to consider the training as an online algorithm with bounded memory (parameters) and utilize results from online learning literature to provide a rigorous exponential lower bound (Lyu et al., 2023; Kol et al., 2017). Hahn & Rofin (2024) shows that while parity is easy to represent using Transformers, the sensitivity structure of the function will require the representation to have large weight norms. A recent work (Abbe et al., 2024a) conjectured that a distribution is weakly learnable by a Transformer if and only if it has constant globality, which is supported by empirical experiments. The work also highlights the parity problem as a notable special case.

**Chain-of-Thoughts (CoT).** CoT is the technique to let the model generate a reasoning process before final answers. CoT prompting has proven effective in enhancing language models' reasoning capabilities (Wei et al., 2022; Zhang et al., 2022; Wang et al., 2023a; Zhou et al., 2023b; Wang & Zhou, 2024; Creswell et al., 2023). Training with CoT data has further improved the model's capability in performing complex reasoning (see Qwen (2024); Yue et al. (2023); Yu et al. (2024); Kim et al. (2023) and reference therein). Different lines of work have explored the effect of CoT. From the representation theory perspective, works including Feng et al. (2024); Merrill & Sabharwal (2024); Li et al. (2024d); Nowak et al. (2024); Wen et al. (2024) show that CoT can provably expand the representation power of neural architectures including Transformers and RNNs. From the statistical approximation level, prompting with CoT reduces the statistical error (Hu et al., 2024; Prystawski et al., 2023). Li et al. (2023a) studies how MLP learns with CoT data. Li et al. (2024a) studies how Transformers learn to perform CoT through prompting using gradient descent. We differ from this work as (1) we focus on how Transformer captures the sparse sequential dependency in CoT, which is not reflected in their data modeling; (2) we study zero-shot CoT data directly. Dutta et al. (2024); Wang et al. (2024a) studies pre-trained Transformers' activation on CoT data. They highlight the attention head's role in moving the essential information from the context to the reasoning process, which is consistent with our theoretical insight. Concurrently with our work, Kim & Suzuki (2024a) theoretically demonstrated that CoT reduces the number of gradient steps required to solve the parity problem. While their analysis focuses on GD with a noisy oracle, our work investigates mini-batch SGD and provides theoretical guarantees on the exponential reduction in sample complexity.

We defer other related works to Appendix A.

# 6 CONCLUSION AND FUTURE WORKS.

Our work demonstrates that Transformers trained on Chain-of-Thought (CoT) data with sparse sequential dependencies can efficiently learn sparse attention mechanisms, accurately capturing these dependencies while requiring exponentially fewer samples than models trained without CoT. Our current analysis of CoT training assumes that each subsequent token depends on exactly one token in the context and focuses solely on the parity function. A promising future direction is to explore more general scenarios where each token depends on multiple previous tokens and extends to function classes beyond parity.

ACKNOWLEDGEMENTS

Jingzhao Zhang is supported by National Key R&D Program of China 2024YFA1015800 and Shanghai Qi Zhi Institute Innovation Program.

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

## A    ADDITIONAL RELATED WORK

**Transformer Optimization Dynamics.** Our works fall in the line of works that studies the optimization dynamics of Transformers on synthetic datasets (Li et al., 2023b; Chen et al., 2024; Kim & Suzuki, 2024b; Wibisono & Wang; Chan et al., 2022; Wibisono & Wang, 2024; Cole et al., 2024; Sheen et al., 2024; Chen et al., 2024; Tian et al., 2023; Nichani et al., 2024; Li et al., 2024b). Similar to our works, Wang et al. (2024b) highlights that Transformers can learn to select sparse critical tokens from the context on linear data. We differ from their work in studying the mini-batch optimization dynamics on nonlinear data and hence establishing sample complexity bounds.

## B    OMITTED PROOF

### B.1    NOTATION AND ASSUMPTIONS.

We will denote a sequence of binary variables as $\mathbf{b}[1], \ldots, \mathbf{b}[T]$ with $\mathbf{b}[i] \in \{0, 1\}$. We will use $[j]$ to denote variables corresponding to the $j-$th token dimension without otherwise specified.

#### B.1.1    TRAINING SPECIFICATION

**Training Update**    We will consider the following 1-pass SGD with batch size $B$:

$$L^{(t)} = \sum_{s=1}^{B} \sum_{i=n+1}^{n+k} \ell(\mathcal{T}[w^{(t)}](\mathbf{b}^{(t,s)})[i], \mathbf{b}^{(t,s)}[i+1]).$$

$$w^{(t+1)} = w^{(t)} - \eta_t \frac{1}{B} \nabla_w L^{(t)}.$$

Here $w$ includes $A$ and $W$. We will leave $h$ unchanged for the simplicity of analysis.

### B.2    REPRESENTATION THEORY

*Proof of Theorem 1.* We will set $d = \Theta(k \log(n/\delta))$ and $m = k + 1$. We will use $p(x, y)$ as shorthand for $2(x - 1) + y + 1$.

Define embedding matrix $M$ as follows

$$M_{p(i,b)} = e_{S[i],b}^T.$$

for $\{S[1], \ldots, S[p]\} = \mathcal{S}$. Then $M \in \mathbb{R}^{2k \times d}$.

**Lemma 1.** *For large enough $n$, with probability $1 - \delta$, it holds that*

$$\|M^T\|_{\mathrm{op}} = O(1). \quad \lambda_{\min}(MM^T) > 1/12.$$

*Proof.* The first result follows from Lemma 28, noted that $K = \Theta(\sqrt{1/d})$. The second result follows from Lemma 29. □

**Lemma 2.** *With probability $1 - \delta/4$ over the randomness of embedding $e_{j,b}$ for $j \in \mathcal{S}, b \in \{0, 1\}$, there exists $u_{j,b}$, satisfying that $\forall j \in \mathcal{S}, b \in \{0, 1\}$,*

$$\forall j' \in \mathcal{S}, b' \in \{0, 1\}, \langle u_{j,b}, e_{j',b'} \rangle = \mathbf{1}((j, b) = (j', b')).$$

*Further,*

$$\| \sum_{j \in \mathcal{S}, b \in \{0,1\}} u_{j,b}\|_2 = O(\sqrt{k}).$$

*Proof.* We can simply choose

$$u_{S[i],b} = M^T(MM^T)^{-1}o_{2(i-1)+b}.$$

with $o_{2(i-1)+b}$ being the one-hot vector in $\mathbb{R}^{2k}$. □

**Lemma 3.** *When the event in Lemma 2 happens, there exists $v_j$,*

$$\langle v_j, e_{j,0} \rangle = \langle v_j, e_{j,1} \rangle = 1.$$
$$\forall j' \neq j, j \in S, \langle v_{j'}, e_{j,b} \rangle = 0.$$

*Further,*

$$\| \sum_{j \in S} v_j \|_2 \leq O(\sqrt{k}).$$

*Proof.* We can choose $v_j = u_{j,0} + u_{j,1}$. □

**Lemma 4.** *When $v_i$ defined in Lemma 2 exists, for all $j \notin S, b \in \{0,1\}$, with probability $1 - \delta$, it holds that*

$$|\langle \sum_{j' \in S} v_{j'}, e_{j,b} \rangle| \leq 1/2.$$

*Proof.* Noted that $e_{j,b}$ is independent from $v_{j'}$, assuming $V = \sum_{j' \in S} v_{j'}$, then $\|V\|_2 = O(\sqrt{k})$, using Azuma-Hoeffding Bound, with probability $1 - \delta/2$

$$|\langle \sum_{j' \in S} v_{j'}, e_{j,b} \rangle| = O(\frac{\| \sum_{j' \in S} v_{j'} \|_2}{\sqrt{d}} \sqrt{\log(2k/\delta)}) = O(\sqrt{\frac{k \log(2k/\delta)}{d}}) \leq \frac{1}{2}.$$

The proof is then completed. □

Assuming the event in Lemmas 2 and 4 happens, defined $V = \sum_{j' \in S} v_{j'}$ and $U = \sum_{j' \in S} u_{j',1}$

$$A = 40 \log(n) V e_{n+1,0}^T \quad W_{r,1:d} = a_r e_{n+1,0} \quad W_{r,d+1:2d} = b_r U.$$

for some scalar $a_r, b_r$ to be determined.

For an arbitrary **b**, it holds that,

1. If $j \in S$, $e_{j,\mathbf{b}[j]}^T A e_{n+1,0} = 40 \log n$.

2. If $j \notin S$, $e_{j,\mathbf{b}[j]}^T A e_{n+1,0} \leq 20 \log n$.

If we define

$$z_j = e_{j,\mathbf{b}[j]}^T A e_{n+1,0}.$$

The softmax value is given by:

$$\text{softmax}(z)_{j,n+1} = \frac{e^{z_j}}{\sum_{j'} e^{z_{j'}}}.$$

1. For $j \in S$:
$$e^{z_j} = e^{40 \log n} = n^{40}.$$

2. For $j \notin S$:
$$e^{z_j} \leq e^{20 \log n} = n^{20}.$$

With $|S| = k$, the denominator of the softmax becomes:

$$\sum_{j'} e^{z_{j'}} = \sum_{j' \in S} e^{40 \log n} + \sum_{j' \notin S} e^{z_{j'}} \in [kn^{40}, kn^{40} + n^{21}].$$

For $j \in S$:

$$\text{softmax}(z)_{j,n+1} = \frac{n^{40}}{kn^{40} + O(n^{20})} = \frac{1}{k} \cdot \frac{1}{1 + O(n^{-19})} = \frac{1}{k} + O(n^{-19}).$$

For $j \notin \mathcal{S}$:

$$\text{softmax}(z)_{j,n+1} = \frac{e^{z_j}}{kn^{40} + O(n^{20})} \leq \frac{n^{20}}{kn^{40}} = O(n^{-20}).$$

We then have

$$\text{softmax}(\mathbf{E}(\mathbf{b})A\mathbf{E}(\mathbf{b}))[j, n+1] = \frac{\mathbf{1}(j \in \mathcal{S})}{k} + O(\frac{1}{n^7}).$$

As

$$\langle U, e_{j,b} \rangle = \mathbf{1}(j \in \mathcal{S}, b = 1).$$

We have that

$$\langle U, \mathcal{A}(\mathbf{E}(\mathbf{b}))[n+1] \rangle = \frac{1}{k} \sum_{j \in \mathcal{S}} \mathbf{b}[j] + O(\frac{1}{n^6}).$$

Now we only need to map from the summation $\sum_{j \in \mathcal{S}} \mathbf{b}[j]$ to the parity of the summation. We set

$$b_r = 2k,$$
$$a_r = \begin{cases} -4\lceil \frac{r}{2} \rceil + 4 & \text{if } 1 \leq r \leq m, \\ 1 & \text{if } r = m+1, \\ -4\lfloor \frac{r-m}{2} \rfloor + 2 & \text{if } m+2 \leq r \leq 2m. \end{cases}$$
$$h_r = \begin{cases} 1 & \text{if } 1 \leq r \leq m, \\ -1 & \text{if } m+1 \leq r \leq 2m, \end{cases}$$

Then

$$\mathcal{T}(\mathbf{E}(\mathbf{b}))[n+1] = \sum_{r=1}^{k} h_r \text{ReLU}(a_r - b_r \langle U, \mathcal{A}(\mathbf{E}(\mathbf{b}))[n+1] \rangle)$$
$$= \sum_{r=1}^{m} \text{ReLU}(a_r + 2\sum_{j \in S} \mathbf{b}[j]) - \sum_{r=m}^{2m} \text{ReLU}(a_r + 2\sum_{j \in S} \mathbf{b}[j]) + O(\frac{1}{n^4})$$
$$= (-1)^{\sum_{j \in S} \mathbf{b}[j]+1} + O(\frac{1}{n^4})$$
$$= (-1)^{\mathbf{b}[n+2]+1} + O(\frac{1}{n^4}).$$

Thus $\text{sgn}(\mathcal{T}(\mathbf{E}(\mathbf{b}))[n+1]) = (-1)^{\mathbf{b}[n+2]}$ for large enough $n$.

Note that the range of parameters is polynomial with $n$, thus could be represented with $\Theta(\log n)$ precision. With $\Theta(\log n)$ precision, the error of each activation can be bounded by a small inverse polynomial of $n$ and hence we can show the same result.

$\square$

### B.3 DYNAMICS WITHOUT COT

*Proof.* We will utilize the Theorem 6 in Lyu et al. (2023), which shows that any branching programs with $o(nk)$ memory will require exponential samples to learn sparse parities with constant passes. Here the frozen embedding matrix $e$, which will utilize naively $O(nd)$ memory, can't be saved in the memory. However, we can take the alternative approach to regenerate $e$ using a random number generator with a fixed seed on each step. This allow us to simulate standard SGD optimization with $o(nk)$ memory, which is a special case of branching programs. $\square$

### B.4 DYNAMICS WITH COT

**Assumption 2.** *Consider the following conditions for sufficiently large $n$:*

1. *The secret Hamming weight satisfies $k \in \left[ \frac{n}{\log^5(n/\delta)}, \frac{n}{\log^4(n/\delta)} \right]$.*

2. *Set $d = k \log^{1.1}\left(\frac{n}{\delta}\right)$ and $m = 2k$. This implies $md \log n = o(nk)$.*

3. *Define the batch size as $B = C_2 n \log^{20}\left(\frac{n}{\delta}\right)$ for a sufficiently large constant $C_2 = \frac{1.28 \times 10^7}{\epsilon^2}$.*

4. *Set the learning rates to be*

$$\eta_0 = \eta_1 = \frac{m\epsilon\sqrt{B}}{100 \log\left(\frac{n}{\delta}\right)}, \quad \eta_2 = \frac{4k\epsilon}{3}.$$

**Lemma 5.** *Under Assumption 2, the following conditions hold as $n \to \infty$:*

1. $\lim_{n\to\infty} \frac{d}{n} = 0$ *and* $\lim_{n\to\infty} \frac{B}{nk} = 0$.

2. $\lim_{n\to\infty} \sqrt{\frac{2k}{d} \log\left(\frac{Bmn}{\delta}\right)} = 0$.

3. *Each of the following expressions tends to zero:*

   (a) $\frac{\sqrt{k \log\left(\frac{300mnB}{\delta}\right)}}{\sqrt{B}}$.

   (b) $\frac{\sqrt{2k \log\left(\frac{400mn}{\delta}\right)}}{\sqrt{d}}$.

   (c) $\frac{\sqrt{2nk \log\left(\frac{300mnB}{\delta}\right)}}{\sqrt{Bd}}$.

4. $\eta_0 = \eta_1 \leq \min\left\{ \frac{3m\epsilon\sqrt{B}}{80\sqrt{\log\left(\frac{300nB}{\delta}\right)}}, \frac{mn\epsilon}{120} \right\}$.

5. $\lim_{n\to\infty} \log^2\left(\frac{300mnB}{\delta}\right) \frac{n\sqrt{nk}}{d\sqrt{Bd}} = 0$.

6. $\frac{\eta_0\eta_1}{256mn^2} > 40 \log n$.

7. $\lim_{n\to\infty} \frac{\eta_2}{\eta_0} = 0$.

8. $\lim_{n\to\infty} (\eta_0 + \eta_1) < 5n\eta_2$.

*Proof.* The conditions in Lemma 5 are satisfied based on the definitions provided in Assumption 2. Below, we outline the verification for each condition:

1. **Limit of $\frac{d}{n}$ and $\frac{B}{nk}$:**

$$\frac{d}{n} = \frac{k \log^{1.1}\left(\frac{n}{\delta}\right)}{n} \in \left[ \frac{1}{\log^{3.9}(n/\delta)}, \frac{1}{\log^{2.9}(n/\delta)} \right] \to 0 \text{ as } n \to \infty.$$

   Similarly,

$$\frac{B}{nk} = \frac{C_2 n \log^{20}\left(\frac{n}{\delta}\right)}{nk} = \frac{C_2 \log^{20}\left(\frac{n}{\delta}\right)}{k} \leq \frac{C_2 \log^{20}\left(\frac{n}{\delta}\right)}{\frac{n}{\log^5(n/\delta)}} = C_2 \frac{\log^{25}(n/\delta)}{n} \to 0.$$

2. **Limit of $\sqrt{\frac{2k}{d} \log\left(\frac{Bmn}{\delta}\right)}$:**

$$\sqrt{\frac{2k}{d} \log\left(\frac{Bmn}{\delta}\right)} = \sqrt{\frac{2k}{k \log^{1.1}\left(\frac{n}{\delta}\right)} \log\left(C_2 n \log^{20}\left(\frac{n}{\delta}\right) \cdot mn/\delta\right)}.$$

Simplifying,

$$\sqrt{\frac{2k}{d} \log\left(\frac{Bmn}{\delta}\right)} = \sqrt{\frac{2}{\log^{1.1}\left(\frac{n}{\delta}\right)} \cdot O(\log n)} = O\left(\frac{\sqrt{\log n}}{\log^{0.55}(n/\delta)}\right) \to 0.$$

3. **Limits of Sub-Inequalities (a), (b), and (c)**:

(a)

$$\frac{\sqrt{k \log\left(\frac{300mnB}{\delta}\right)}}{\sqrt{B}} = \frac{\sqrt{k \cdot O(\log n)}}{\sqrt{C_2 n \log^{20}\left(\frac{n}{\delta}\right)}} = O\left(\frac{\sqrt{k} \cdot \sqrt{\log n}}{\sqrt{n} \cdot \log^{10}(n/\delta)}\right).$$

Given $k \le \frac{n}{\log^2(n/\delta)}$,

$$\frac{\sqrt{k} \cdot \sqrt{\log n}}{\sqrt{n} \cdot \log^{10}(n/\delta)} \le \frac{\sqrt{\frac{n}{\log^2(n/\delta)}} \sqrt{\log n}}{\sqrt{n} \cdot \log^{10}(n/\delta)} = \frac{\log^{0.5}(n)}{\log^{11}(n/\delta)} \to 0.$$

(b)

$$\frac{\sqrt{2k \log\left(\frac{400mn}{\delta}\right)}}{\sqrt{d}} = \frac{\sqrt{2k \cdot O(\log n)}}{\sqrt{k \log^{1.1}\left(\frac{n}{\delta}\right)}} = O\left(\frac{\sqrt{\log n}}{\log^{0.55}(n/\delta)}\right) \to 0.$$

(c)

$$\frac{\sqrt{2nk \log\left(\frac{300mnB}{\delta}\right)}}{\sqrt{Bd}} = \frac{\sqrt{2nk \cdot O(\log n)}}{\sqrt{C_2 n \log^{20}\left(\frac{n}{\delta}\right) \cdot k \log^{1.1}\left(\frac{n}{\delta}\right)}} = O\left(\frac{\sqrt{nk \log n}}{\sqrt{nk} \cdot \log^{10.55}(n/\delta)}\right)$$

$$= O\left(\frac{\sqrt{\log n}}{\log^{10.55}(n/\delta)}\right) \to 0.$$

4. **Bound on $\eta_0$ and $\eta_1$**:

$$\eta_0 = \eta_1 = \frac{m\epsilon\sqrt{B}}{100 \log\left(\frac{n}{\delta}\right)} = \frac{2k\epsilon\sqrt{C_2 n \log^{20}\left(\frac{n}{\delta}\right)}}{100 \log\left(\frac{n}{\delta}\right)} = O\left(k\epsilon\sqrt{n} \log^9(n/\delta)\right).$$

Comparing to the minimum of the two terms:

$$\frac{3m\epsilon\sqrt{B}}{80\sqrt{\log\left(\frac{300nB}{\delta}\right)}} = O\left(k\epsilon\sqrt{n} \log^{10}(n/\delta)\right),$$

and

$$\frac{mn\epsilon}{120} = O(kn\epsilon).$$

Since $\eta_0$ scales similarly to the first term and $\sqrt{n} \log^{10}(n/\delta) \ll n$ for large $n$, the inequality $\eta_0 \le \min\{\cdot\}$ holds.

5. **Limit of $\log^2\left(\frac{300mnB}{\delta}\right) \frac{n\sqrt{nk}}{d\sqrt{Bd}}$**:

$$\log^2\left(\frac{300mnB}{\delta}\right) = O(\log^2(n/\delta)),$$

and

$$\frac{n\sqrt{nk}}{d\sqrt{Bd}} = \frac{n\sqrt{nk}}{k \log^{1.1}\left(\frac{n}{\delta}\right) \cdot \sqrt{C_2 n \log^{20}\left(\frac{n}{\delta}\right) \cdot k \log^{1.1}\left(\frac{n}{\delta}\right)}}$$

$$= O\left(\frac{n\sqrt{nk}}{k \log^{1.1}(n/\delta)\sqrt{nk} \log^{10.55}(n/\delta)}\right) = O\left(\frac{n}{k \log^{11.65}(n/\delta)}\right).$$

Given $k \geq \frac{n}{\log^5(n/\delta)}$,

$$\frac{n}{k \log^{11.65}(n/\delta)} \leq \frac{n}{\frac{n}{\log^5(n/\delta)} \log^{11.65}(n/\delta)} = \frac{1}{\log^{5.65}(n/\delta)}.$$

Therefore,

$$\log^2\left(\frac{300mnB}{\delta}\right)\frac{n\sqrt{nk}}{d\sqrt{Bd}} = O\left(\frac{\log^2(n/\delta)}{\log^{5.65}(n/\delta)}\right) = O\left(\log^{-4.65}(n/\delta)\right) \to 0 \text{ as } n \to \infty.$$

6. **Limit of** $\frac{\eta_0\eta_1}{256mn^2}$: As

$$\eta_0\eta_1 = \left(\frac{m\epsilon\sqrt{B}}{100\log\left(\frac{n}{\delta}\right)}\right)^2 = \frac{m^2\epsilon^2 B}{10^4 \log^2\left(\frac{n}{\delta}\right)}.$$

Substituting into the left term:

$$\frac{\eta_0\eta_1}{256\,m\,n^2} = \frac{m^2\epsilon^2 B}{256 \times 10^4 \log^2\left(\frac{n}{\delta}\right)\,m\,n^2} = \frac{m\epsilon^2 B}{256 \times 10^4 \log^2\left(\frac{n}{\delta}\right)\,n^2}.$$

Substitute $m = 2k$ and $B = C_2 n \log^{10}\left(\frac{n}{\delta}\right)$:

$$\frac{\eta_0\eta_1}{256\,m\,n^2} = \frac{2k\epsilon^2 C_2 n \log^{10}\left(\frac{n}{\delta}\right)}{256 \times 10^4 \log^2\left(\frac{n}{\delta}\right) n^2} = \frac{2kC_2 \log^8\left(\frac{n}{\delta}\right)}{256 \times 10^4 n}.$$

Setting $C_2 = \frac{1.28\times10^7}{\epsilon^2}$:

$$\frac{\eta_0\eta_1}{256\,m\,n^2} \geq \frac{2k \times 1.28 \times 10^7 \log^8\left(\frac{n}{\delta}\right)}{256 \times 10^4 \times n} = 10 \times \frac{k \log^8\left(\frac{n}{\delta}\right)}{n}.$$

Using $k \geq \frac{n}{\log^5\left(\frac{n}{\delta}\right)}$ from the assumption:

$$\frac{\eta_0\eta_1}{256\,m\,n^2} \geq 10 \times \frac{\frac{n}{\log^5\left(\frac{n}{\delta}\right)} \log^8\left(\frac{n}{\delta}\right)}{n} = 10\log^3\left(\frac{n}{\delta}\right).$$

Since $\log^3\left(\frac{n}{\delta}\right) > 4\log n$ for sufficiently large $n$, we have:

$$10\log^3\left(\frac{n}{\delta}\right) > 40\log n.$$

Thus,

$$\frac{\eta_0\eta_1}{256\,m\,n^2} > 40\log n.$$

7. We aim to show that:

$$\frac{\eta_2}{\eta_0} \to 0 \quad \text{as} \quad n \to \infty.$$

Here

$$\eta_0 = \frac{2k\epsilon\sqrt{C_2\,n\,\log^{10}\left(\frac{n}{\delta}\right)}}{100\log\left(\frac{n}{\delta}\right)} = \frac{2k\epsilon\sqrt{C_2}\sqrt{n}\log^5\left(\frac{n}{\delta}\right)}{100\log\left(\frac{n}{\delta}\right)} = \frac{k\epsilon\sqrt{C_2}\sqrt{n}\log^4\left(\frac{n}{\delta}\right)}{50}.$$

Thus, the ratio $\frac{\eta_2}{\eta_0}$ is:

$$\frac{\eta_2}{\eta_0} = \frac{\frac{4k\epsilon}{3}}{\frac{k\epsilon\sqrt{C_2}\sqrt{n}\log^4\left(\frac{n}{\delta}\right)}{50}} = \frac{4k\epsilon \times 50}{3k\epsilon\sqrt{C_2}\sqrt{n}\log^4\left(\frac{n}{\delta}\right)} = \frac{200}{3\sqrt{C_2}\sqrt{n}\log^4\left(\frac{n}{\delta}\right)}.$$

Substituting $C_2 = \frac{1.28 \times 10^7}{\epsilon^2}$:

$$\frac{\eta_2}{\eta_0} = \frac{200}{3\sqrt{\frac{1.28 \times 10^7}{\epsilon^2}}\sqrt{n}\log^4\left(\frac{n}{\delta}\right)} = \frac{200\epsilon}{3 \times 3580\sqrt{n}\log^4\left(\frac{n}{\delta}\right)}.$$

As $n \to \infty$, $\sqrt{n}\log^4\left(\frac{n}{\delta}\right) \to \infty$, hence:

$$\frac{\eta_2}{\eta_0} \to 0.$$

8. **Limit of** $(\eta_0 + \eta_1) < 5n\eta_2$:

$$\eta_0 + \eta_1 = 2 \cdot \frac{m\epsilon\sqrt{B}}{100\log\left(\frac{n}{\delta}\right)} = O\left(\frac{m\epsilon\sqrt{B}}{\log\left(\frac{n}{\delta}\right)}\right).$$

$$5n\eta_2 = 5n \cdot \frac{4k\epsilon}{3} = O(nk\epsilon).$$

Given $\sqrt{B} = O\left(\sqrt{n}\log^{10}(n/\delta)\right)$,

$$\frac{m\epsilon\sqrt{B}}{\log\left(\frac{n}{\delta}\right)} = O\left(\frac{k\epsilon\sqrt{n}\log^{10}(n/\delta)}{\log(n/\delta)}\right) = O\left(k\epsilon\sqrt{n}\log^9(n/\delta)\right).$$

Since $nk\epsilon$ grows faster than $k\epsilon\sqrt{n}\log^9(n/\delta)$ for large $n$, the inequality $(\eta_0 + \eta_1) < 5n\eta_2$ holds.

This concludes the proof. □

*Proof.* We will set our hyperparameters according to Assumption 2. We will first warmup our analysis on linear loss $\ell(\hat{y}, y) = (-1)^y\hat{y}$. We can rewrite the training dynamics in this case as

$$w^{(t+1)} = w^{(t)} - \eta_t\frac{1}{B}\sum_{s=1}^{B}\sum_{i=n+1}^{n+k}\nabla_w\ell(\mathcal{T}[w^{(t)}](\mathbf{b}^{(t,s)})[i], \mathbf{b}^{(t,s)}[i+1])$$

$$= w^{(t)} - \eta_t\frac{1}{B}\sum_{s=1}^{B}\sum_{i=n+1}^{n+k}(-1)^{\mathbf{b}^{(t,s)}[i+1]}\nabla_w\mathcal{T}[w^{(t)}](\mathbf{b}^{(t,s)})[i]. \tag{2}$$

We will train the model end to end in three steps and analyze the evolution of all the weight changes. The final results can be shown combining Lemmas 21 and 26. Lemma 27 generalizes the analysis to hinge loss. It is also easy to verify that all the results hold if the parameters has only $O(\log n)$ precision. □

### B.4.1 AUXILIARY STATISTICS

To simplify our calculations, we define several auxiliary statistics on the data. Table 1 provides the definitions and rough orders of these statistics, including logarithmic terms.

**Lemma 6.** *Fixing $j, b_1, b_2$, let $\mathcal{F}_{(i,t,s)}$ be the filtration generated by the random variables $\{\boldsymbol{b}^{(t,s)}[i] \mid \max\{n + 1, j\} \leq i, S[i] \neq j, t' \leq t, s \leq B\}$ ordered in lexicographic order, the process*

$$(-1)^{\boldsymbol{b}^{(t,s)}[i+1]}\mathbf{1}(\boldsymbol{b}^{(t,s)}[j] = b_1, \boldsymbol{b}^{(t,s)}[i] = b_2)$$

*is a martingale with respect to $\{\mathcal{F}_t\}_t$.*

*Proof.* $\mathbf{b}[i+1] = \mathbf{b}[i] \oplus \mathbf{b}[S[i]]$, and since $S[i] \neq j$, $\mathbf{b}[S[i]]$ is independent from $\mathbf{b}[j], \ldots, \mathbf{b}[i]$. Thus,

$$\mathbb{E}[(-1)^{\mathbf{b}^{(t,s)}[i+1]} \mid \mathcal{F}_{(i,t,s)}] = 0,$$

provided $\mathbf{1}(\mathbf{b}^{(t+1,s)}[j] = b_1)$ is constant. Hence, this process is a martingale. □

| Statistic | Definition | Rough Order |
|---|---|---|
| $\delta_{t,i,b}$ | $\displaystyle\sum_{s=1}^{B}(-1)^{\mathbf{b}^{(t,s)}[i+1]}\mathbf{1}\left(\mathbf{b}^{(t,s)}[i]=b\right)$ | $\mathcal{O}\left(\sqrt{B\log\left(\frac{nB}{\delta}\right)}\right)$ |
| $\alpha_{t,i,j,b_1,b_2}$ | $\displaystyle\sum_{s=1}^{B}(-1)^{\mathbf{b}^{(t,s)}[i+1]}\mathbf{1}\left(\mathbf{b}^{(t,s)}[j]=b_1,\mathbf{b}^{(t,s)}[i]=b_2\right)$ | $\bar{\alpha}_{i,j,b_1,b_2}+$ $\mathcal{O}\left(\sqrt{B\log\left(\frac{nB}{\delta}\right)}\right)$ |
| $\bar{\alpha}_{i,j,b_1,b_2}$ | $\begin{cases}\frac{(-1)^{b_1+b_2}\mathbf{1}(j=S[i])B}{4}, & i\geq n+2\\ \frac{(-1)^{b_1}\mathbf{1}(j=S[i],\ b_2=0)B}{2}, & i=n+1\end{cases}$ | $\mathcal{O}(B)$ when $j=S[i]$, 0 otherwise |
| $\beta_{t,j,b_1}$ | $\displaystyle\sum_{\substack{i=\max\{n+1,j\}\\ S[i]\neq j}}^{n+k}\sum_{b_2=0}^{1}\frac{1}{i}\alpha_{t,i,j,b_1,b_2}$ | $\mathcal{O}\left(\frac{\sqrt{Bk\log\left(\frac{nB}{\delta}\right)}}{n}\right)$ |
| $\psi_{t,r,j,b_1}$ | $\displaystyle\sum_{\substack{i=\max\{n+1,j\}\\ S[i]\neq j}}^{n+k}\sum_{b_2=0}^{1}\frac{1}{i}\mathbf{1}\left(\nu_{r,i,b_2}>0\right)\alpha_{t,i,j,b_1,b_2}$ | $\mathcal{O}\left(\frac{\sqrt{Bk\log\left(\frac{mnB}{\delta}\right)}}{n}\right)$ |
| $\gamma_{i,b,i',b_2}$ | $\displaystyle\frac{1}{i'}\left(\sum_{r=1}^{2m}\mathbf{1}\left(\nu_{r,i,b}>0\right)\mathbf{1}\left(\nu_{r,i',b_2}>0\right)\right)$ $-\frac{1}{i'}\frac{1+\mathbf{1}\left(i=i',\ b=b_2\right)}{2}m$ | $\mathcal{O}\left(\frac{\sqrt{m\log\left(\frac{n}{\delta}\right)}}{n}\right)$ |
| $\zeta_{t,i,b,j,b_1}$ | $\displaystyle\sum_{r=1}^{2m}\mathbf{1}\left(\nu_{r,i,b}>0\right)\psi_{t,r,j,b_1}$ | $\mathcal{O}\left(\frac{m\sqrt{kB\log\left(\frac{mnB}{\delta}\right)}}{n}\right)$ |

Table 1: Definitions and rough orders of auxiliary statistics, including logarithmic terms. Here, $t\in\{0,1,2\}$, $i'\in[n+1,n+k+1]$, $i,j\in[n+k]$, $b,b_1,b_2\in\{0,1\}$, $s\in[B]$, and $r\in[2m]$.

This version maintains the essential details while focusing on the core aspects of the martingale property and independence conditions.

**Lemma 7.** *For each batch, the data inside the batch is approximately balanced. With probability at least $1-\delta/10$, the following bounds hold for all $t\in[3]$, $i,j\in[n+k]$, $b,b_1,b_2\in\{0,1\}$, and $r\in[m]$ for statistics defined in Table 1:*

$$|\delta_{t,i,b}|\leq\sqrt{B\log\left(\frac{300nB}{\delta}\right)}$$

$$|\alpha_{t,i,j,b_1,b_2}-\bar{\alpha}_{i,j,b_1,b_2}|\leq\sqrt{2B\log\left(\frac{300nB}{\delta}\right)}\mathbf{1}\left(j=S[i]\right)$$

$$|\beta_{t,j,b_1}|\leq\frac{\sqrt{2Bk\log\left(\frac{300nB}{\delta}\right)}}{n}$$

$$|\psi_{t,r,j,b_1}|\leq\frac{2\sqrt{Bk\log\left(\frac{300mnB}{\delta}\right)}}{n}$$

$$|\gamma_{i,b,i',b_2}|\leq\frac{\sqrt{2m\log\left(\frac{200n}{\delta}\right)}}{n}$$

$$|\zeta_{t,i,b,j,b_1}|\leq\frac{4m\sqrt{kB\log\left(\frac{300mnB}{\delta}\right)}}{n}$$

*Proof.* Fix any $t$, $i$, and $b$. For each $s\in[B]$, define $X_s=(-1)^{\mathbf{b}^{(t,s)}[i+1]}\mathbf{1}\left(\mathbf{b}^{(t,s)}[i]=b\right)$. The sequence $\{X_s\}_{s=1}^{B}$ consists of independent random variables with $|X_s|\leq1$ and zero mean. By the

Azuma-Hoeffding inequality,

$$\Pr\left(|\delta_{t,i,b}| > R\right) \le 2\exp\left(-\frac{R^2}{2B}\right).$$

Setting $R = \sqrt{2B\log\left(\frac{300nB}{\delta}\right)}$, we get

$$\Pr\left(|\delta_{t,i,b}| > R\right) \le \frac{\delta}{150nB}.$$

Applying a union bound over all $t$, $i$, and $b$, we ensure that with probability at least $1 - \delta/50$, the bound on $\delta_{t,i,b}$ holds for all choices.

Similarly, for $\alpha_{t,i,j,b_1,b_2}$, we fix $t$, $i$, $j$, $b_1$, and $b_2$, and define

$$Y_s = (-1)^{\mathbf{b}^{(t,s)}[i+1]}\mathbf{1}\left(\mathbf{b}^{(t,s)}[j] = b_1, \mathbf{b}^{(t,s)}[i] = b_2\right).$$

Again, $\{Y_s\}_{s=1}^B$ are independent with $|Y_s| \le 1$ and mean $\bar{\alpha}_{i,j,b_1,b_2}/B$. Applying the Azuma-Hoeffding inequality as before, we obtain the stated bound for $\alpha_{t,i,j,b_1,b_2}$.

For $\beta_{t,j,b_1}$ and $\psi_{t,r,j,b_1}$, the bound can be derived similarly combining Lemma 6 and Azuma-Hoeffding bound.

For $\gamma_{i,b,i',b_2}$, consider the sum $\sum_{r=1}^{2m} \mathbf{1}\left(\nu_{r,i,b} > 0\right)\mathbf{1}\left(\nu_{r,i',b_2} > 0\right)$. Each term is an independent Bernoulli random variable with mean $\frac{1}{2}\left(1 + \mathbf{1}\left(i = i', b = b_2\right)\right)$. The variance of the sum is bounded by $m$. Applying the Azuma-Hoeffding inequality, we get

$$\Pr\left(|\gamma_{i,b,i',b_2}| > R\right) \le 2\exp\left(-\frac{R^2 n^2}{2m}\right).$$

Setting $R = \frac{\sqrt{2m\log\left(\frac{200n}{\delta}\right)}}{n}$, we obtain the desired bound.

For $\zeta_{t,i,b,j,b_1}$, we note that it is a sum over $2m$ terms, each involving $\psi_{t,r,j,b_1}$, which are bounded as in Lemma 7. The total number of terms is $2m$, and each term is bounded by $\frac{2\sqrt{Bk\log\left(\frac{300mnB}{\delta}\right)}}{n}$. $\quad\square$

### B.4.2 FIRST STEP: CONFIGURING THE MLPS

**Lemma 8.** *For the dynamics following Equation* (2)*, attention weight stays constant in the first step, i.e.,* $A^{(1)} = A^{(0)} = 0$.

*Proof.* Because $W_{:,d+1:2d} = 0$, $\nabla_A \mathcal{T}[w^{(0)}](\mathbf{b})[i] = 0$. $\quad\square$

**Lemma 9.** *For the dynamics following Equation* (2)*, attention output stays constant in the first step, and*

$$\forall t \in \{0,1\}, i \in [n+k+1], \mathcal{A}[A^{(t)}](\mathbf{E}(\boldsymbol{b}))[i] = \frac{1}{i}\sum_{j=1}^{i} e_{i,\boldsymbol{b}[i]}.$$

*Proof.* This follows from the definition of the attention module and Lemma 8. $\quad\square$

**Lemma 10.** *For an input $\boldsymbol{b}$, With probability $1 - \frac{\delta}{50Bm}$ , at initialization, whether a neuron $r$ outputs nonzero value at position $i$ is determined by $\nu_{r,i,\boldsymbol{b}[i]}$.*

$$\mathbf{1}\left(\langle W_{r,1:d}^0, \mathbf{E}(\boldsymbol{b})[i]\rangle > 0\right) = \mathbf{1}\left(\nu_{r,i,\boldsymbol{b}[i]} > 0\right).$$

*Further,*

$$\left|\langle W_{r,1:d}^0, \mathbf{E}(\boldsymbol{b})[i]\rangle - \nu_{r,i,\boldsymbol{b}[i]}\right| < \epsilon/100.$$

*Proof.* Under Assumption 1,

$$W_{r,1:d}^0 = \sum_{i=n+1}^{n+k} \sum_{b=0}^{1} \nu_{r,i,b} e_{i,b}$$

Because,

$$\sum_{i=n+1}^{n+k} \sum_{b=0}^{1} \nu_{r,i,b}^2 = 2k\epsilon^2.$$

By Lemma 30, with probability $1 - \frac{\delta}{50Bm}$,

$$\left| \langle W_{r,1:d}^0, \mathbf{E}(\mathbf{b})[i] \rangle - \nu_{r,i,\mathbf{b}[i]} \right| \leq \sqrt{\frac{4\log(\frac{100Bm}{\delta})k}{d}} \epsilon$$

By Lemma 5.2, $\sqrt{\frac{4k}{d} \log(100Bm/\delta)} < 0.01$, we can conclude that

$$\Pr\left( \left| \sum_{i'=n+1}^{n+k} \sum_{l=1}^{d} \nu_{r,i',b'} \mathbf{1}(i' \neq i) e_{i',\mathbf{b}[i']}[l] e_{i,\mathbf{b}[i]}[l] \right| \geq \epsilon/100 \right) \leq \frac{\delta}{50Bm}.$$

The proof is then complete. $\qquad\square$

**Lemma 11.** *With probability $1 - \frac{\delta}{50}$, the following $\mathcal{E}_1$ happens: for any input $\mathbf{b}^{(0,s)}$ in the first batch, at initialization, for any $r$, whether a neuron $r$ outputs nonzero value at position $i$ is determined by $\nu_{r,i,\mathbf{b}[i]}$.*

$$\mathbf{1}\left( \langle W_{r,1:d}^0, \mathbf{E}(\mathbf{b})[i] \rangle > 0 \right) = \mathbf{1}\left( \nu_{r,i,\mathbf{b}[i]} > 0 \right).$$

*Further,*

$$\left| \langle W_{r,1:d}^0, \mathbf{E}(\mathbf{b})[i] \rangle - \nu_{r,i,\mathbf{b}[i]} \right| < \epsilon/100.$$

*Proof.* Apply union bound to Lemma 10 over neuron dimension and data in the first batch. $\qquad\square$

**Lemma 12.** *When $\mathcal{E}_1$ defined in Lemma 11 happens, for all $i' \in [n+1, n+k+1], b' \in \{0,1\}$, the gradient on $W_{r,1:d}$ satisfies that,*

$$\frac{d\mathcal{L}^{(0)}}{dW_{r,1:d}} = \frac{h_r}{B} \sum_{i=n+1}^{n+k} \sum_{b=0}^{1} \mathbf{1}(\nu_{r,i,b} > 0)\delta_{0,i,b} e_{i,b}.$$

*Proof.*

$$\begin{aligned}
\frac{d\mathcal{L}^{(0)}}{dW_{r,1:d}} &= \frac{1}{B} \sum_{s=1}^{B} \sum_{i=n+1}^{n+k} (-1)^{\mathbf{b}^{(0,s)}[i+1]} \nabla_{W_{r,1:d}} \mathcal{T}[w^{(t)}](\mathbf{b}^{(0,s)})[i] \\
&= \frac{h_r}{B} \sum_{s=1}^{B} \sum_{i=n+1}^{n+k} (-1)^{\mathbf{b}^{(0,s)}[i+1]} e_{i,\mathbf{b}^{(t,s)}[i]} \mathbf{1}(\nu_{r,i,\mathbf{b}^{(0,s)}[i]} > 0).
\end{aligned}$$

We can then expand this formula to have

$$\sum_{s=1}^{B}\sum_{i=n+1}^{n+k}(-1)^{\mathbf{b}^{(0,s)}[i+1]}e_{i,\mathbf{b}^{(0,s)}[i]}\mathbf{1}(\nu_{r,i,\mathbf{b}^{(0,s)}[i]}>0)$$

$$=\sum_{s=1}^{B}\sum_{i=n+1}^{n+k}\sum_{b=0}^{1}(-1)^{\mathbf{b}^{(0,s)}[i+1]}\mathbf{1}(\mathbf{b}^{(0,s)}[i]=b)e_{i,b}\mathbf{1}(\nu_{r,i,b}>0)$$

$$=\sum_{i=n+1}^{n+k}\sum_{b=0}^{1}e_{i,b}\mathbf{1}(\nu_{r,i,b}>0)\sum_{s=1}^{B}(-1)^{\mathbf{b}^{(0,s)}[i+1]}\mathbf{1}(\mathbf{b}^{(0,s)}[i]=b)$$

$$=\sum_{i=n+1}^{n+k}\sum_{b=0}^{1}e_{i,b}\mathbf{1}(\nu_{r,i,b}>0)\delta_{0,i,b}.$$

The proof is then complete. $\qquad\square$

**Lemma 13.** *With probability $1-0.13\delta$, for all $i'\in[n+1,n+k+1],b'\in\{0,1\}$, the gradient on $W_{r,1:d}$ satisfies that,*

$$\left|\left\langle\frac{d\mathcal{L}^{(0)}}{dW_{r,1:d}},e_{i,b}\right\rangle\right|\le\frac{\sqrt{\log(300nB/\delta)}}{m\sqrt{B}}.$$

*Proof.* Combining Lemmas 12 and 30, we have that for any $r$, with probability $1-\delta/100m$,

$$\left|\left\langle\frac{h_r}{B}\sum_{i=n+1}^{n+k}\sum_{b=0}^{1}\mathbf{1}(\nu_{r,i,b}>0)\delta_{0,i,b}e_{i,b},e_{i,b}\right\rangle\right|\le\frac{1}{2mB}\sup\delta_{0,i,b}\left(1+\sqrt{\frac{2k\log\left(\frac{200m}{\delta}\right)}{d}}\right)$$

By Lemmas 5 and 7, with probability $1-11\delta/100$, for all $r$, it holds that for all $r,i',b'$

$$\left|\left\langle\frac{h_r}{B}\sum_{i=n+1}^{n+k}\sum_{b=0}^{1}\mathbf{1}(\nu_{r,i,b}>0)\delta_{0,i,b}e_{i,b},e_{i,b}\right\rangle\right|<\frac{1}{mB}\sup\delta_{0,i,b}<\frac{\sqrt{\log(300nB/\delta)}}{m\sqrt{B}}.$$

Combining with Lemma 12 and applying union bound concludes the proof. $\qquad\square$

**Lemma 14.** *For $\bar{\alpha}$ and $\psi$ defined in Appendix B.4.1, When $\mathcal{E}_1$ defined in Lemma 11 happens, for all $i'\in[n+1,n+k+1],b'\in\{0,1\}$, the gradient on $W_{r,d+1:2d}$ satisfies that,*

$$\frac{d\mathcal{L}^{(0)}}{dW_{r,d+1:2d}}=\frac{h_r}{B}\sum_{i=n+1}^{n+k}\sum_{b_1=0}^{1}\sum_{b_2=0}^{1}\frac{1}{i}\mathbf{1}(\nu_{r,i,b_2}>0)\bar{\alpha}_{i,S[i],b_1,b_2}e_{S[i],b_1}+\frac{h_r}{B}\sum_{j=1}^{n+k}\sum_{b_1=0}^{1}\psi_{0,r,j,b_1}e_{j,b_1}.$$

*Proof.* With probability $1-\delta/50$, $\mathcal{E}_1$ defined in Lemma 11 happens, and

$$\frac{d\mathcal{L}^{(0)}}{dW_{r,d+1:2d}}=\frac{1}{B}\sum_{s=1}^{B}\sum_{i=n+1}^{n+k}(-1)^{\mathbf{b}^{(0,s)}[i+1]}\nabla_{W_{r,1:d}}\mathcal{T}[w^{(t)}](\mathbf{b}^{(0,s)})[i]$$

$$=\frac{h_r}{B}\sum_{s=1}^{B}\sum_{i=n+1}^{n+k}(-1)^{\mathbf{b}^{(0,s)}[i+1]}\mathcal{A}[A^{(t)}](\mathbf{E}(\mathbf{b}^{(0,s)}))[i]\mathbf{1}(\nu_{r,i,\mathbf{b}^{(0,s)}[i]}>0)$$

$$=\frac{h_r}{B}\sum_{s=1}^{B}\sum_{i=n+1}^{n+k}(-1)^{\mathbf{b}^{(0,s)}[i+1]}\mathbf{1}(\nu_{r,i,\mathbf{b}^{(0,s)}[i]}>0)\frac{1}{i}\sum_{j=1}^{i}\sum_{b_1=0}^{1}\mathbf{1}\left(\mathbf{b}[j]=b\right)e_{j,b_1}$$

Rearranging the summation, and use $\alpha$ defined in Lemma 7,,

$$\frac{d\mathcal{L}^{(0)}}{dW_{r,d+1:2d}}=\frac{h_r}{B}\sum_{j=1}^{n+k}\sum_{b_1=0}^{1}e_{j,b_1}\sum_{i=\max\{n+1,j\}}^{n+k}\sum_{b_2=0}^{1}\frac{1}{i}\mathbf{1}(\nu_{r,i,b_2}>0)\alpha_{0,i,j,b_1,b_2}.$$

Breaking the summation,

$$\frac{d\mathcal{L}^{(0)}}{dW_{r,d+1:2d}} = \frac{h_r}{B} \sum_{i=n+1}^{n+k} \sum_{b_1=0}^{1} \sum_{b_2=0}^{1} \frac{1}{i} \mathbf{1}(\nu_{r,i,b_2} > 0)\alpha_{0,i,S[i],b_1,b_2} e_{S[i],b_1}$$

$$+ \frac{h_r}{B} \sum_{j=1}^{n+k} \sum_{b_1=0}^{1} e_{j,b_1} \sum_{i \in [\max\{n+1,j\}.n+k], S[i] \neq j} \sum_{b_2=0}^{1} \frac{1}{i} \mathbf{1}(\nu_{r,i,b_2} > 0)\alpha_{0,i,j,b_1,b_2}$$

This simplifies to

$$\frac{d\mathcal{L}^{(0)}}{dW_{r,d+1:2d}} = \frac{h_r}{B} \sum_{i=n+1}^{n+k} \sum_{b_1=0}^{1} \sum_{b_2=0}^{1} \frac{1}{i} \mathbf{1}(\nu_{r,i,b_2} > 0)\bar{\alpha}_{i,S[i],b_1,b_2} e_{S[i],b_1} + \frac{h_r}{B} \sum_{j=1}^{n+k} \sum_{b_1=0}^{1} \psi_{0,r,j,b_1} e_{j,b_1}.$$

The proof is then complete. $\square$

This leads to the following bound on the gradient,

**Lemma 15.** *With probability* $1 - 0.16\delta$, *for all* $i \in [n+k+1], b \in \{0,1\}$, *the gradient on* $W_{r,d+1:2d}$ *satisfies that,*

$$\left| \left\langle \frac{d\mathcal{L}^{(0)}}{dW_{r,d+1:2d}}, e_{i,b} \right\rangle \right| \leq \frac{3}{2mn}.$$

*Proof.* Denote

$$G_r = \frac{h_r}{B} \sum_{i=n+1}^{n+k} \sum_{b_1=0}^{1} \sum_{b_2=0}^{1} \frac{1}{i} \mathbf{1}(\nu_{r,i,b_2} > 0)\bar{\alpha}_{i,S[i],b_1,b_2} e_{S[i],b_1} + \frac{h_r}{B} \sum_{j=1}^{n+k} \sum_{b_1=0}^{1} \psi_{0,r,j,b_1} e_{j,b_1}.$$

By Lemma 30, for all $r \in [2m], i \in [n+k+1], b \in \{0,1\}$, with probability $1 - \delta/25$,

$$|\langle G_r, e_{i,b} \rangle| \leq \frac{1}{2mB} \left| \sum_{i'=n+1}^{n+k} \sum_{b_2=0}^{1} \frac{1}{i} \mathbf{1}(\nu_{r,i,b_2} > 0)\bar{\alpha}_{i,S[i],b,b_2} \mathbf{1}(S[i'] = i) + \psi_{0,r,i,b} \right|$$

$$+ \frac{\sqrt{2\log(\frac{400mn}{\delta})}}{2mB\sqrt{d}} \left( \sqrt{\sum_{i=n+1}^{n+k} \sum_{b_1=0}^{1} \sum_{b_2=0}^{1} \frac{1}{i^2} \mathbf{1}(\nu_{r,i,b_2} > 0)\bar{\alpha}^2_{i,S[i],b_1,b_2}} + \sqrt{\sum_{j=1}^{n+k} \sum_{b_1=0}^{1} \psi^2_{0,r,j,b_1}} \right)$$

By Lemma 7, we have with probability $1 - \delta/10$,

$$\sup \psi_{0,r,i,b} \leq \frac{2\sqrt{Bk\log\left(\frac{300mnB}{\delta}\right)}}{n}, \quad \sup \bar{\alpha}_{i,S[i],b_1,b_2} \leq B/2.$$

Combining with Lemma 5,

$$\frac{1}{2mB} \left| \sum_{i'=n+1}^{n+k} \sum_{b_2=0}^{1} \frac{1}{i} \mathbf{1}(\nu_{r,i,b_2} > 0)\bar{\alpha}_{i,S[i],b,b_2} \mathbf{1}(S[i'] = i) \right| \leq \frac{1}{2mn}$$

$$\frac{1}{2mB} |\psi_{0,r,i,b}| \leq \frac{2\sqrt{k\log\left(\frac{300mnB}{\delta}\right)}}{mn\sqrt{B}} \leq \frac{1}{200mn}$$

$$\frac{\sqrt{2\log(\frac{400mn}{\delta})}}{2mB\sqrt{d}} \sqrt{\sum_{i=n+1}^{n+k} \sum_{b_1=0}^{1} \sum_{b_2=0}^{1} \frac{1}{i^2} \mathbf{1}(\nu_{r,i,b_2} > 0)\bar{\alpha}^2_{i,S[i],b_1,b_2}} \leq \frac{\sqrt{2k\log(\frac{400mn}{\delta})}}{2mn\sqrt{d}} \leq \frac{1}{200mn},$$

$$\frac{\sqrt{2\log(\frac{400mn}{\delta})}}{2mB\sqrt{d}} \sqrt{\sum_{j=1}^{n+k} \sum_{b_1=0}^{1} \psi^2_{0,r,j,b_1}} \leq \frac{2\sqrt{2nk}\log(\frac{300mnB}{\delta})}{mn\sqrt{Bd}} \leq \frac{1}{200mn}.$$

Further by Lemma 14, we have that $\frac{d\mathcal{L}^{(0)}}{dW_{r,d+1:2d}} = G_r$ with probability $1 - \delta/50$. This concludes the proof. $\square$

B.4.3 SECOND STEP: CONFIGURING THE ATTENTION

**Lemma 16.** *With probability $1 - 0.29\delta$, the following $\mathcal{E}_2$ happens: the change in the neuron outcome is small after the first step for any input $\boldsymbol{b}$,*

$$\sup_r \left| \langle W_r^1 - W_r^0, [\mathbf{E}(\boldsymbol{b})[i], \mathcal{A}[A^1](\mathbf{E}(\boldsymbol{b}))[i]] \rangle \right| < 3\epsilon/80.$$

*This leads to the case that for any input $\boldsymbol{b}^{(1,s)}$ in the second batch, for any $r$, whether a neuron $r$ outputs nonzero value at position $i$ is determined by $\nu_{r,i,\boldsymbol{b}[i]}$.*

$$\mathbf{1}\left( \langle W_{r,1:d}^1, \mathbf{E}(\boldsymbol{b})[i] \rangle > 0 \right) = \mathbf{1}\left( \nu_{r,i,\boldsymbol{b}[i]} > 0 \right).$$

*Proof.* By Lemma 13, with probability $1 - 0.13\delta$,

$$\sup_r \left| \langle W_{r,1:d}^1 - W_{r,1:d}^0, \mathbf{E}(\mathbf{b})[i] \rangle \right| = \eta_0 \left| \left\langle \frac{d\mathcal{L}^{(0)}}{dW_{r,1:d}}, \mathbf{E}(\mathbf{b})[i] \right\rangle \right| \leq \eta_0 \frac{\sqrt{\log(300nB/\delta)}}{m\sqrt{B}}.$$

By Lemma 15, with probability $1 - 0.16\delta$,

$$\sup_r \left| \langle W_{r,d+1:2d}^1 - W_{r,d+1:2d}^0, \mathcal{A}[A^1](\mathbf{E}(\mathbf{b}))[i] \rangle \right| = \eta_0 \left| \left\langle \frac{d\mathcal{L}^{(0)}}{dW_{r,d+1:2d}}, \mathcal{A}[A^1](\mathbf{E}(\mathbf{b}))[i] \right\rangle \right| \leq \eta_0 \frac{3}{2mn}.$$

Hence, as $\eta_0 \leq \min\{ \frac{3m\epsilon\sqrt{B}}{80\sqrt{\log(300nB/\delta)}}, \frac{mn\epsilon}{120} \}$ (Lemma 5.3), we conclude that

$$\sup_r \left| \langle W_r^1 - W_r^0, [\mathbf{E}(\mathbf{b})[i], \mathcal{A}[A^1](\mathbf{E}(\mathbf{b}))[i]] \rangle \right| < 3\epsilon/80.$$

The proof is then complete. $\square$

This shows that the gradient of MLP on the second and first steps is the same in distribution.

**Lemma 17.** *With probability $1 - 0.58\delta$, inequalities in Lemma 7 hold and for $t \in \{0,1\}, \forall i \in [n+k], \forall b \in \{0,1\}$,*

$$\frac{d\mathcal{L}^{(t)}}{dW_{r,1:d}} = \frac{h_r}{B} \sum_{i=n+1}^{n+k} \sum_{b=0}^{1} \mathbf{1}(\nu_{r,i,b} > 0)\delta_{t,i,b}e_{i,b}.$$

$$\frac{d\mathcal{L}^{(t)}}{dW_{r,d+1:2d}} = \frac{h_r}{B} \sum_{i=n+1}^{n+k} \sum_{b_1=0}^{1} \sum_{b_2=0}^{1} \frac{1}{i}\mathbf{1}(\nu_{r,i,b_2} > 0)\bar{\alpha}_{i,S[i],b_1,b_2}e_{S[i],b_1} + \frac{h_r}{B} \sum_{j=1}^{n+k} \sum_{b_1=0}^{1} \psi_{t,r,j,b_1}e_{j,b_1}.$$

$$\left| \left\langle \frac{d\mathcal{L}^{(t)}}{dW_{r,d+1:2d}}, e_{i,b} \right\rangle \right| \leq \frac{3}{2mn}.$$

*Moreover for any input $\boldsymbol{b}$,*

$$\sup_r \left| \langle W_r^t - W_r^0, [\mathbf{E}(\boldsymbol{b})[i], \mathcal{A}[A^1](\mathbf{E}(\boldsymbol{b}))[i]] \rangle \right| < 3\epsilon/40.$$

*Further,*

$$\mathbf{1}\left( \left\langle W_{r,1:d}^t, \mathbf{E}(\boldsymbol{b}^{(t,s)})[i] \right\rangle > 0 \right) = \mathbf{1}\left( \nu_{r,i,\boldsymbol{b}^{(t,s)}[i]} > 0 \right).$$

*Proof.* Consider the backward calculation of MLP, it is decided by the input and the corresponding activation of neurons, which jointly follows the same distribution in the first and second steps. The proof is similar to Lemmas 12, 14 and 16. $\square$

We will now show that while MLP changes little, it is enough to provide high-quality gradient information to inform the attention layer. To this end, we will define the following terms,

$$\kappa_{t,i,b} = -\sum_{r=1}^{2m} \mathbf{1}\left( \nu_{r,i,b} > 0 \right) h_r W_{r,d+1:2d}^{(t)}$$

$$\Delta_{t,i,b,j,b_1} = \langle \kappa_{t,i,b}, e_{j,b_1} \rangle$$

Intuitively, the FFN with weight $W^{(1)}$ maps attention output $x$ to $-\kappa_{i,\mathbf{b}[i]}x$ at position $i$.

**Lemma 18.** *For $\alpha, \beta, \gamma, \zeta$ defined in Appendix B.4.1, when the event in Lemma 17 happens, for any $i$ and $b$,*

$$\kappa_{1,i,b} = \frac{\eta_0}{8Bmi} \sum_{b_1=0}^{1} \bar{\alpha}_{i,S[i],b_1,b} e_{S[i],b_1} + \frac{\eta_0}{4Bm^2} \left( \sum_{i'=n+1}^{n+k} \sum_{b_1=0}^{1} \sum_{b_2=0}^{1} \gamma_{i,b,i',b_2} \bar{\alpha}_{i',S[i'],b_1,b_2} e_{S[i'],b_1} \right)$$

$$+ \frac{\eta_0}{4Bm^2} \left( \sum_{j=1}^{n+k} \sum_{b_1=0}^{1} \zeta_{0,i,b,j,b_1} e_{j,b_1} \right).$$

*Proof.* When the event in Lemma 17 happens, we have that

$$\frac{d\mathcal{L}^{(0)}}{dW_{r,d+1:2d}} = \frac{h_r}{B} \sum_{i'=n+1}^{n+k} \sum_{b_1=0}^{1} \sum_{b_2=0}^{1} \frac{1}{i'} \mathbf{1}(\nu_{r,i',b_2} > 0) \bar{\alpha}_{i',S[i'],b_1,b_2} e_{S[i'],b_1} + \frac{h_r}{B} \sum_{j=1}^{n+k} \sum_{b_1=0}^{1} \psi_{0,r,j,b_1} e_{j,b_1}.$$

Summing over the axis of $r$ as in $\kappa$,

$$\kappa_{1,i,b} = \frac{\eta_0}{4Bm^2} \left( \sum_{i'=n+1}^{n+k} \sum_{b_1=0}^{1} \sum_{b_2=0}^{1} \left( \sum_{r=1}^{2m} \mathbf{1}\left(\nu_{r,i,b} > 0\right) \mathbf{1}\left(\nu_{r,i',b_2} > 0\right) \right) \frac{1}{i'} \bar{\alpha}_{i',S[i'],b_1,b_2} e_{S[i'],b_1} \right)$$

$$+ \frac{\eta_0}{4Bm^2} \left( \sum_{j=1}^{n+k} \sum_{b_1=0}^{1} e_{j,b_1} \left( \sum_{r=1}^{2m} \mathbf{1}\left(\nu_{r,i,b} > 0\right) \psi_{0,r,j,b_1} \right) \right).$$

Recall that $\gamma_{i,b,i',b_2} = \frac{1}{i'} \left( \sum_{r=1}^{2m} \mathbf{1}\left(\nu_{r,i,b} > 0\right) \mathbf{1}\left(\nu_{r,i',b_2} > 0\right) \right) - \frac{1}{i'} \frac{1+\mathbf{1}\left(i=i',\ b=b_2\right)}{2} m$. We then have

$$\kappa_{1,i,b} = \frac{\eta_0}{4Bm} \left( \sum_{i'=n+1}^{n+k} \sum_{b_1=0}^{1} \sum_{b_2=0}^{1} \frac{1}{i'} \frac{1+\mathbf{1}\left(i=i',\ b=b_2\right)}{2} \bar{\alpha}_{i',S[i'],b_1,b_2} e_{S[i'],b_1} \right)$$

$$+ \frac{\eta_0}{4Bm^2} \left( \sum_{i'=n+1}^{n+k} \sum_{b_1=0}^{1} \sum_{b_2=0}^{1} \gamma_{i,b,i',b_2} \bar{\alpha}_{i',S[i'],b_1,b_2} e_{S[i'],b_1} \right)$$

$$+ \frac{\eta_0}{4Bm^2} \left( \sum_{j=1}^{n+k} \sum_{b_1=0}^{1} e_{j,b_1} \left( \sum_{r=1}^{2m} \mathbf{1}\left(\nu_{r,i,b} > 0\right) \psi_{0,r,j,b_1} \right) \right).$$

The first term can be greatly simplified, as

$$\alpha_{i',S[i'],b_1,0} + \alpha_{i',S[i'],b_1,1} = 0.$$

This concludes the proof. $\qquad\square$

**Lemma 19.** *With probability $1 - 0.6\delta$, inequalities and equalities in Lemmas 7, 17 and 18 hold, and for all $i, b_2, j, b_1$,*

$$\left| \Delta_{1,i,b_2,j,b_1} - \frac{\eta_0}{8Bmi} \bar{\alpha}_{i,S[i],b_1,b_2} \mathbf{1}(j = S[i]) \right| \le \frac{10\eta_0}{mn} \frac{\sqrt{nk}}{\sqrt{Bd}} \log(300mnB/\delta).$$

*Proof.* Define

$$H_{i,b} = \frac{\eta_0}{8Bmi} \sum_{b_1=0}^{1} \bar{\alpha}_{i,S[i],b_1,b} e_{S[i],b_1}$$

$$+ \frac{\eta_0}{4Bm^2} \left( \sum_{i'=n+1}^{n+k} \sum_{b_1=0}^{1} \sum_{b_2=0}^{1} \gamma_{i,b,i',b_2} \bar{\alpha}_{i',S[i'],b_1,b_2} e_{S[i'],b_1} \right)$$

$$+ \frac{\eta_0}{4Bm^2} \left( \sum_{j=1}^{n+k} \sum_{b_1=0}^{1} \zeta_{i,b,j,b_1} e_{j,b_1} \right)$$

As the inequalities in Lemma 7 hold, by Lemmas 5 and 30, with probability $1 - \delta/50$, it holds that,

$$\left| \frac{\eta_0}{4Bm^2} \left\langle \left( \sum_{i'=n+1}^{n+k} \sum_{b_1=0}^{1} \sum_{b_2=0}^{1} \gamma_{i,b,i',b_2} \bar{\alpha}_{i',S[i'],b_1,b_2} e_{S[i'],b_1} \right), e_{j,b_1} \right\rangle \right|$$

$$\leq \frac{\eta_0 \sup \bar{\alpha} \sup \gamma}{4Bm^2} \left( 1 + 4\sqrt{\frac{k \log(200n/\delta)}{d}} \right)$$

$$\leq \frac{\eta_0}{4mn} \sqrt{\frac{2k \log\left(\frac{200n}{\delta}\right)}{md}}.$$

and,

$$\left| \frac{\eta_0}{4Bm^2} \left\langle \left( \sum_{j=1}^{n+k} \sum_{b_1=0}^{1} \zeta_{i,b,j,b_1} e_{j,b_1} \right), e_{j,b_1} \right\rangle \right|$$

$$\leq \frac{\eta_0 \sup \zeta}{4Bm^2} \left( 1 + 4\sqrt{\frac{n \log(200n/\delta)}{d}} \right)$$

$$\leq \frac{\eta_0}{mn} \left( \sqrt{\frac{k}{B}} \log(200n/\delta) + 4\sqrt{\frac{nk}{dB}} \log(200n/\delta) \right).$$

and,

$$\left| \frac{\eta_0}{8Bmi} \left\langle \left( \sum_{b_1=0}^{1} \bar{\alpha}_{i,S[i],b_1,b} e_{S[i],b_1} \right), e_{j,b_1} \right\rangle - \frac{\eta_0}{8Bmi} \bar{\alpha}_{i,S[i],b_1,b} \mathbf{1}(j = S[i]) \right| \leq \frac{2\sqrt{\log(200n/\delta)}}{8mn\sqrt{d}}.$$

With $d < n$ and $B < nk$ (Lemma 5), we conclude that,

$$\left| \langle H_{i,b}, e_{j,b_1} \rangle - \frac{\eta_0}{8Bmi} \bar{\alpha}_{i,S[i],b_1,b} \mathbf{1}(j = S[i]) \right| \leq \frac{10\eta_0}{mn} \frac{\sqrt{nk}}{\sqrt{Bd}} \log(300mnB/\delta).$$

By Lemma 18, we have the desired result. $\qquad \square$

**Lemma 20.** *Under the setting of Lemma 19, for every $\boldsymbol{b} \in \{\boldsymbol{b}^{(1,s)} \mid s \in [B]\}$,*

$$\frac{d\ell(\mathcal{T}(\boldsymbol{b})[i], \boldsymbol{b}[i+1])}{dA} = (-1)^{\boldsymbol{b}[i+1]+1} \sum_{j=1}^{i} \langle \kappa_{1,i,\boldsymbol{b}[i]}, e_{j,\boldsymbol{b}[j]} \rangle \frac{\partial \text{softmax}\left( \mathbf{E}(\boldsymbol{b})^T A \mathbf{E}(\boldsymbol{b}) \right)[j,i]}{\partial A},$$

*with*

$$\frac{\partial \text{softmax}\left( \mathbf{E}(\boldsymbol{b})^T A \mathbf{E}(\boldsymbol{b}) \right)[j,i]}{\partial A} = \text{softmax}(Z)[j,i] \cdot \left( \mathbf{E}(\boldsymbol{b})[j] - \sum_{p=1}^{i} \text{softmax}(Z)[p,i]\mathbf{E}(\boldsymbol{b})[p] \right) \mathbf{E}(\boldsymbol{b})[i]^T$$

*Proof.* Our goal is to calculate the gradient of the loss with respect to the attention layer. We will first calculate the gradient of the loss with respect to the output of the attention layer, and then use the chain rule. When the event in Lemma 17 happens, for any binary sequence $\mathbf{b}$ in the second batch, we have that for any $i$,

$$\frac{d\mathcal{T}(\mathbf{b})[i]}{d\mathcal{A}(\mathbf{E}(\mathbf{b}))[i]} = \sum_{r=1}^{2m} h_r W_r^{(1)} \mathbf{1}\left( \left( \left( W^{(1)} \begin{bmatrix} \mathbf{E}(\mathbf{b}) \\ \mathcal{A}(\mathbf{E}(\mathbf{b})) \end{bmatrix} \right) [i]_r > 0 \right) \right)$$

$$= \sum_{r=1}^{2m} h_r W_{r,d+1:2d}^{(1)} \mathbf{1}\left( \nu_{r,i,\mathbf{b}[i]} > 0 \right)$$

This then implies that,

$$\frac{d\ell(\mathcal{T}(\mathbf{b})[i], \mathbf{b}[i+1])}{d\mathcal{A}(\mathbf{E}(\mathbf{b}))[i]} = (-1)^{\mathbf{b}[i+1]} \left( \sum_{r=1}^{2m} h_r W_{r,d+1:2d}^{(1)} \mathbf{1}\left( \nu_{r,i,\mathbf{b}[i]} > 0 \right) \right)$$

We now calculate the gradient of $A$ (the attention matrix) to the attention output.

$$\frac{\partial \mathcal{A}(\mathbf{E}(\mathbf{b}))[i]}{\partial A} = \frac{\partial \left( \mathbf{E}(\mathbf{b}) \text{softmax}\left(\mathbf{E}(\mathbf{b})^T A \mathbf{E}(\mathbf{b})\right)\right)[i]}{\partial A}$$

$$= \frac{\sum_{j=1}^{i} \partial \mathbf{E}(\mathbf{b})[j] \text{softmax}\left(\mathbf{E}(\mathbf{b})^T A \mathbf{E}(\mathbf{b})\right)[j,i]}{\partial A}$$

$$= \sum_{j=1}^{i} \mathbf{E}(\mathbf{b})[j] \frac{\partial \text{softmax}\left(\mathbf{E}(\mathbf{b})^T A \mathbf{E}(\mathbf{b})\right)[j,i]}{\partial A}$$

Hence, the gradient of the loss with respect to the attention matrix is

$$\frac{d\ell(\mathcal{T}(\mathbf{b})[i], \mathbf{b}[i+1])}{dA}$$

$$= \sum_{j=1}^{i} \frac{d\ell(\mathcal{T}(\mathbf{b})[i], \mathbf{b}[i+1])}{d\mathcal{A}(\mathbf{E}(\mathbf{b}))[i]} \mathbf{E}(\mathbf{b})[j] \frac{\partial \text{softmax}\left(\mathbf{E}(\mathbf{b})^T A \mathbf{E}(\mathbf{b})\right)[j,i]}{\partial A}$$

$$= (-1)^{\mathbf{b}[i+1]} \sum_{j=1}^{i} \sum_{r=1}^{2m} h_r \mathbf{1}\left(\nu_{r,i,\mathbf{b}[i]} > 0\right) \langle W_{r,d+1:2d}^{(1)}, \mathbf{E}(\mathbf{b})[j]\rangle \frac{\partial \text{softmax}\left(\mathbf{E}(\mathbf{b})^T A \mathbf{E}(\mathbf{b})\right)[j,i]}{\partial A}$$

$$= (-1)^{\mathbf{b}[i+1]+1} \sum_{j=1}^{i} \langle \kappa_{1,i,\mathbf{b}[i]}, e_{j,\mathbf{b}[j]}\rangle \frac{\partial \text{softmax}\left(\mathbf{E}(\mathbf{b})^T A \mathbf{E}(\mathbf{b})\right)[j,i]}{\partial A}$$

We will use $Z$ to denote $\mathbf{E}(\mathbf{b})^T A \mathbf{E}(\mathbf{b})$, and calculate the derivative of th+e softmax function applied to $Z$ with respect to $A$.

$$\frac{\partial \text{softmax}(Z)[j,i]}{\partial A} = \frac{\partial}{\partial A} \left( \frac{e^{Z[j,i]}}{\sum_{p=1}^{i} e^{Z[p,i]}} \right)$$

$$= \frac{e^{Z[j,i]} \cdot \frac{\partial Z[j,i]}{\partial A} \cdot \sum_{p=1}^{i} e^{Z[p,i]} - e^{Z[j,i]} \cdot \sum_{p=1}^{i} e^{Z[p,i]} \cdot \frac{\partial Z[p,i]}{\partial A}}{\left(\sum_{p=1}^{i} e^{Z[p,i]}\right)^2}$$

Given $Z[p,i] = \mathbf{E}(\mathbf{b})[p]^T A \mathbf{E}(\mathbf{b})[i]$, the derivative with respect to $A$ is:

$$\frac{\partial Z[p,i]}{\partial A} = \mathbf{E}(\mathbf{b})[p] \cdot \mathbf{E}(\mathbf{b})[i]^T$$

Substituting the derivatives into the quotient rule expression:

$$\frac{\partial \text{softmax}(Z)[j,i]}{\partial A} = \frac{e^{Z[j,i]} \cdot \mathbf{E}(\mathbf{b})[j]\mathbf{E}(\mathbf{b})[i]^T \cdot \sum_{p=1}^{i} e^{Z[p,i]} - e^{Z[j,i]} \cdot \sum_{p=1}^{i} e^{Z[p,i]} \cdot \mathbf{E}(\mathbf{b})[p]\mathbf{E}(\mathbf{b})[i]^T}{\left(\sum_{p=1}^{i} e^{Z[p,i]}\right)^2}$$

Thus, the derivative is:

$$\frac{\partial \text{softmax}\left(\mathbf{E}(\mathbf{b})^T A \mathbf{E}(\mathbf{b})\right)[j,i]}{\partial A} = \text{softmax}(Z)[j,i] \cdot \left( \mathbf{E}(\mathbf{b})[j] - \sum_{p=1}^{i} \text{softmax}(Z)[p,i] \mathbf{E}(\mathbf{b})[p] \right) \mathbf{E}(\mathbf{b})[i]^T$$

The proof is then complete. $\qquad\square$

**Lemma 21.** *After two training steps, the attention layer will show the following structure. With probability $1 - 0.7\delta$, for all $i, b, i', b_2$,*

1. If $j' = S[i']$, then $\langle e_{j',b_1'}, A^{(2)} e_{i',b_2'}\rangle \geq \frac{\eta_0 \eta_1}{256mn^2}$.

2. If $j' \neq S[i']$, then $|\langle e_{j',b_1'}, A^{(2)} e_{i',b_2'}\rangle| \leq \frac{\eta_0 \eta_1}{512mn^2}$.

*Further, events in Lemmas 17 and 18 hold.*

*Proof.* By Lemma 20,

$$\frac{\partial \text{softmax}\left(\mathbf{E}(\mathbf{b})^T A \mathbf{E}(\mathbf{b})\right)[j,i]}{\partial A} = \frac{1}{i}\left(\mathbf{E}(\mathbf{b})[j] - \sum_{p=1}^{i} \frac{1}{i}\mathbf{E}(\mathbf{b})[p]\right)\mathbf{E}(\mathbf{b})[i]^T$$

We can rewrite the gradient as

$$\frac{d\ell(\mathcal{T}(\mathbf{b})[i], \mathbf{b}[i+1])}{dA} = -\frac{1}{i}(-1)^{\mathbf{b}[i+1]}\left(\sum_{j=1}^{i}\langle\kappa_{i,\mathbf{b}[i]}, \mathbf{E}(\mathbf{b})[j]\rangle\mathbf{E}(\mathbf{b})[j]\right)\mathbf{E}(\mathbf{b})[i]^T$$
$$-\frac{1}{i^2}\sum_{j=1}^{i}\sum_{p=1}^{i}(-1)^{\mathbf{b}[i+1]}\langle\kappa_{i,\mathbf{b}[i]}, \mathbf{E}(\mathbf{b})[j]\rangle\mathbf{E}(\mathbf{b})[p]\mathbf{E}(\mathbf{b})[i]^T$$

If we define

$$\mu_{t,j,i,p,b_1,b_2,b_3} = \left(\sum_{s=1}^{B}(-1)^{\mathbf{b}^{(t,s)}[i+1]}\mathbf{1}(\mathbf{b}^{(t,s)}[j] = b_1, \mathbf{b}^{(t,s)}[i] = b_2, \mathbf{b}^{(t,s)}[p] = b_3)\right).$$

Summing over the second batch, we have that

$$-\frac{dL^{(1)}}{dA} = \frac{1}{B}\sum_{i=n+1}^{n+k}\sum_{j=1}^{i}\sum_{b_1=0}^{1}\sum_{b_2=0}^{1}\frac{1}{i}\alpha_{1,i,j,b_1,b_2}\Delta_{1,i,j,b_1,b_2}e_{j,b_1}e_{i,b_2}^T$$
$$-\frac{1}{B}\sum_{i=n+1}^{n+k}\sum_{j=1}^{i}\sum_{p=1}^{i}\sum_{b_1=0}^{1}\sum_{b_2=1}^{1}\sum_{b_3=0}^{1}\frac{1}{i^2}\mu_{1,j,i,p,b_1,b_2,b_3}\Delta_{1,i,j,b_1,b_2}e_{p,b_3}e_{i,b_2}^T.$$

This implies $A^{(2)}$ is updated as,

$$A^{(2)} = -\eta_1 \frac{dL^{(1)}}{dA}$$
$$= \frac{\eta_1}{B}\sum_{i=n+1}^{n+k}\sum_{j=1}^{i}\sum_{b_1=0}^{1}\sum_{b_2=0}^{1}\frac{1}{i}\alpha_{1,i,j,b_1,b_2}\Delta_{1,i,j,b_1,b_2}e_{j,b_1}e_{i,b_2}^T$$
$$- \frac{\eta_1}{B}\sum_{i=n+1}^{n+k}\sum_{j=1}^{i}\sum_{p=1}^{i}\sum_{b_1=0}^{1}\sum_{b_2=1}^{1}\sum_{b_3=0}^{1}\frac{1}{i^2}\mu_{1,j,i,p,b_1,b_2,b_3}\Delta_{1,i,j,b_1,b_2}e_{p,b_3}e_{i,b_2}^T.$$

Recall that our goal is to calculate $e_{j',b_1'}^T A^{(2)} e_{i',b_2'}$. We can then calculate the contribution from each term by separating the calculation as follows:

$$T_j = \frac{\eta_1}{B} \sum_{i\in[\max\{n+1,j\},n+k],b_2\in\{0,1\},S[i]\neq j} \sum_{b_1=0}^{1} \frac{1}{i}\alpha_{1,i,j,b_1,b_2}\Delta_{1,i,j,b_1,b_2}\langle e_{j',b_1'},e_{j,b_1}\rangle\langle e_{i,b_2},e_{i',b_2'}\rangle.$$

$$R = \frac{\eta_1}{B} \sum_{i=n+1}^{n+k} \sum_{b_1=0}^{1}\sum_{b_2=0}^{1} \frac{1}{i}\alpha_{1,i,S[i],b_1,b_2}\Delta_{1,i,S[i],b_1,b_2}\langle e_{j',b_1'},e_{S[i],b_1}\rangle\langle e_{i,b_2},e_{i',b_2'}\rangle.$$

$$U_{j,p} = \frac{\eta_1}{B} \sum_{i=\max\{n+1,j,p\},S[i]\notin\{j,p\}}^{n+k} \sum_{b_1=0}^{1}\sum_{b_2=0}^{1}\sum_{b_3=0}^{1} \frac{1}{i^2}\mu_{1,j,i,p,b_1,b_2,b_3}\Delta_{1,i,j,b_1,b_2}\langle e_{j',b_1'},e_{j,b_1}\rangle\langle e_{p,b_3},e_{i',b_2'}\rangle$$

$$V_p = \frac{\eta_1}{B} \sum_{i=\max\{n+1,p\}}^{n+k} \sum_{b_1=0}^{1}\sum_{b_2=0}^{1} \frac{1}{i^2}\mu_{1,S[i],i,p,b_1,b_2,b_3}\Delta_{1,i,S[i],b_1,b_2}\langle e_{j',b_1'},e_{S[i],b_1}\rangle\langle e_{p,b_3},e_{i',b_2'}\rangle$$

$$W_j = \frac{\eta_1}{B} \sum_{i=\max\{n+1,j\}}^{n+k} \sum_{b_1=0}^{1}\sum_{b_2=0}^{1}\sum_{b_3=0}^{1} \frac{1}{i^2}\mu_{1,j,i,S[i],b_1,b_2,b_3}\Delta_{1,i,j,b_1,b_2}\langle e_{j',b_1'},e_{j,b_1}\rangle\langle e_{p,b_3},e_{i',b_2'}\rangle.$$

$$Y = \frac{\eta_1}{B} \sum_{i=n+1}^{n+k} \sum_{b_1=0}^{1}\sum_{b_2=0}^{1}\sum_{b_3=0}^{1} \frac{1}{i^2}\mu_{1,S[i],i,S[i],b_1,b_2,b_3}\Delta_{1,i,S[i],b_1,b_2}\langle e_{j',b_1'},e_{S[i],b_1}\rangle\langle e_{S[i],b_3},e_{i',b_2'}\rangle.$$

Then we have that

$$e_{j',b_1'}^T A^{(2)} e_{i',b_2'} = \sum_{j=1}^{n+k} T_j + R - \sum_{j=1}^{n+k}\sum_{p=1}^{n+k} U_{j,p} - \sum_{j=1}^{n+k} W_j - \sum_{p=1}^{n+k} V_p + Y.$$

.By Lemma 31, with probability $1 - \delta/100$,

$$\left|\langle e_{i,b_2},e_{i',b_2'}\rangle - \mathbf{1}(i=i'\&b_2=b_2')\right| \leq 2\sqrt{\frac{\log(100nd/\delta)}{d}}.$$

We will also assume the event in Lemma 19 holds.

We will now discuss each term separately

1. For $T_j$, by Lemma 6 and Azuma-Hoeffding bound, with probability $1 - \delta/50$,

$$\left|\sum_{i\in[\max\{n+1,j\},n+k],b_2\in\{0,1\},S[i]\neq j} \sum_{b_2=0}^{1} \frac{1}{i}\alpha_{1,i,j,b_1,b_2}\Delta_{1,i,j,b_1,b_2}\langle e_{j',b_1'},e_{j,b_1}\rangle\langle e_{i,b_2},e_{i',b_2'}\rangle\right|$$

$$\leq 2\sum_{b_1=0}^{1}\sqrt{B\log(\frac{100n}{\delta}) \sum_{i\in[\max\{n+1,j\},n+k],b_2\in\{0,1\},S[i]\neq j} \frac{1}{i^2}\Delta_{1,1,i,j,b_1,b_2}^2\left(\langle e_{j',b_1'},e_{j,b_1}\rangle\right)^2\left(\langle e_{i,b_2},e_{i',b_2'}\rangle\right)^2}$$

$$\leq \frac{2\sqrt{B\log(\frac{100n}{\delta})}}{n}\sup_{j\neq S[i]}|\Delta_{1,i,j,b_1,b_2}|\sum_{b_1=0}^{1}\left|\langle e_{j',b_1'},e_{j,b_1}\rangle\right|\sqrt{\sum_{i\in[\max\{n+1,j\},n+k],b_2\in\{0,1\},S[i]\neq j}\left(\langle e_{i,b_2},e_{i',b_2'}\rangle\right)^2}$$

Now by Lemma 19, we have that

$$\sup_{j\neq S[i]}|\Delta_{1,i,j,b_1,b_2}| \leq \frac{10\eta_0}{mn}\frac{\sqrt{nk}}{\sqrt{Bd}}\log(300mnB/\delta).$$

Further by Lemma 31,

$$\sum_{i\in[\max\{n+1,j\},n+k],b_2\in\{0,1\},S[i]\neq j}\left(\langle e_{i,b_2},e_{i',b_2'}\rangle\right)^2 \leq \frac{8k\log(100nd/\delta)}{d} + 1 \leq 4.$$

$$\left|\langle e_{j',b_1'},e_{j,b_1}\rangle\right| \leq \mathbf{1}(j=j'\&b_1'=b_1) + 2\sqrt{\frac{\log(100nd/\delta)}{d}}.$$

$$|T_j| \leq \frac{40\eta_1\eta_0}{mn^2}\frac{\sqrt{nk}}{B\sqrt{d}}\log^{1.5}(300mnB/\delta)\left(4\sqrt{\frac{\log(100nd/\delta)}{d}} + \mathbf{1}(j = j')\right).$$

Summing over $j$, by Lemma 5, we have that the contribution is bounded by

$$\sum_j |T_j| \leq \frac{160\eta_1\eta_0}{mn^2}\log^2(300mnB/\delta)\frac{n\sqrt{nk}}{d\sqrt{Bd}} \leq \frac{\eta_1\eta_0}{2000mn^2}. \tag{3}$$

2. For $R$, we will directly put everything to an upper bound. We will discuss three cases,

- $\nexists i'', S[i''] = j$, by Lemma 19,

$$|R| \leq \frac{16k\eta_1\log(100nd/\delta)}{Bnd}\sup\alpha\Delta + \frac{\eta_1}{B}\frac{1}{i'}\alpha_{1,i',S[i'],b_1,b_2'}\Delta_{1,i',S[i'],b_1,b_2'}|\langle e_{j',b_1'}, e_{S[i],b_1}\rangle|$$

$$\leq \frac{16k\eta_1\log(100nd/\delta)}{Bnd}\frac{B}{2}\frac{\eta_0}{4mn} + \frac{\eta_1}{Bn}B\frac{\eta_0}{4mn}\sqrt{\frac{\log(100nd/\delta)}{d}}$$

$$= \frac{\eta_0\eta_1 k\log(100nd/\delta)}{mn^2 d} + \frac{\eta_0\eta_1}{4mn^2}\sqrt{\frac{\log(100nd/\delta)}{d}} \leq \frac{\eta_0\eta_1}{2000mn^2}.$$

- $\exists i'', S[i''] = j, i'' \neq i'$, we can get a similar bound,

$$|R| \leq \frac{\eta_0\eta_1}{2000mn^2}.$$

- $S[i'] = j'$, in this case, we can show that,

$$\left|R - \frac{\eta_0\eta_1}{8B^2 mi'}\bar{\alpha}^2_{i,S[i'],b_1,b_2}\right| \leq \frac{\eta_0\eta_1}{2000mn^2}.$$

This concludes that

$$\left|R - \frac{\eta_0\eta_1}{8B^2 mi'}\bar{\alpha}^2_{i,S[i'],b_1,b_2}\mathbf{1}(S[i'] = j')\right| \leq \frac{\eta_0\eta_1}{2000mn^2}. \tag{4}$$

3. $U_{j,p}$ can be bounded in the same way as $T_j$ and we have that,

$$\sum_j\sum_p U_{j,p} \leq \frac{\eta_1\eta_0}{1000mn^2}. \tag{5}$$

4. For $V_p, W_j, Y$, we will directly put everything to an upper bound similar to the bound of $R$, we have that

$$\sum_{p=1}^{n+k}|V_p| \leq \frac{\eta_1\eta_0}{1000mn^2}. \tag{6}$$

$$\sum_{j=1}^{n+k}|W_j| \leq \frac{\eta_1\eta_0}{1000mn^2} \tag{7}$$

$$|Y| \leq \frac{\eta_1\eta_0}{1000mn^2}. \tag{8}$$

Summing over Equations (3) to (8), we have that

$$\left|\left\langle e_{j',b_1'}, A^{(2)}e_{i',b_2'}\right\rangle - \frac{\eta_0\eta_1}{8Bmi'}\bar{\alpha}^2_{i,S[i'],b_1,b_2}\mathbf{1}(S[i'] = j')\right| \leq \frac{\eta_1\eta_0}{512mn^2}.$$

Further

$$\frac{\eta_1}{8B^2 mi'}\bar{\alpha}^2_{i,S[i'],b_1,b_2} \geq \frac{\eta_0\eta_1}{128mn^2}.$$

Hence, we can show that

1. If $j' = S[i']$, then $\langle e_{j',b'_1}, A^{(2)} e_{i',b'_2} \rangle \geq \frac{\eta_0 \eta_1}{256mn^2}$.

2. If $j' \neq S[i']$, then $|\langle e_{j',b'_1}, A^{(2)} e_{i',b'_2} \rangle| \leq \frac{\eta_0 \eta_1}{512mn^2}$.

The proof is complete. $\qquad\square$

**Lemma 22.** *Under the setting of Lemma 21, the attention output is approximately one-hot after the second step, with*

$$\left| \mathcal{A}[A^2](\mathbf{E}(\boldsymbol{b}))[i] - e_{S[i],\boldsymbol{b}[S[i]]} \right| < 1/n^{10}.$$

*Proof of Lemma 22.* This is a direct combination with Lemmas 5, 21 and 32. $\qquad\square$

### B.4.4 THIRD STEP: MOVING MLP IN LINEAR REGIME

**Lemma 23.** *With probability $1 - 0.8\delta$, the gradient of the FFN layer on the third step can be written as,*

$$\left| \left\langle \frac{d\mathcal{L}^{(2)}}{dW_{r,1:d}}, e_{i,b} \right\rangle \right| \leq \frac{\sqrt{\log(300nB/\delta)}}{m\sqrt{B}}.$$

$$\frac{d\mathcal{L}^{(2)}}{dW_{r,d+1:2d}} = h_r \sum_{i=n+1}^{n+k} \sum_{b_1=0}^{1} \sum_{b_2=0}^{1} \frac{1}{i} \mathbf{1}(\nu_{r,i,b_2} > 0)(-1)^{b_1+b_2} e_{S[i],b_1} + O(\frac{1}{n^8}).$$

*Further the event in Lemmas 21 and 22 hold.*

*Proof.* The proof is similar to Lemmas 12 and 14, switching the original output with the near one-hot output of the attention layer. $\qquad\square$

**Lemma 24.** *With probability $1 - 0.9\delta$, the attention output is almost unchanged after the third step, with*

$$\left| \mathcal{A}[A^3](\mathbf{E}(\boldsymbol{b}))[i] - e_{S[i],\boldsymbol{b}[S[i]]} \right| < 1/n^9.$$

*Further, events in Lemma 23 hold.*

*Proof.* By Lemmas 5, 20 and 22, $A^{(3)} - A^{(2)}$ is of order $1/n^8$, this implies that the attention weight and output is almost unchanged. $\qquad\square$

**Lemma 25.** *With probability $1 - 0.91\delta$,*

$$\left| \left\langle W_{r,d+1:2d}^{(2)}, \mathcal{A}(A^3)[\mathbf{E}(\boldsymbol{b})][i] \right\rangle + \text{sign}(\nu_{r,i,\boldsymbol{b}[S[i]]} h_r) \frac{2\epsilon}{3} \mathbf{1}(\nu_{r,i,0}\nu_{r,i,1} < 0) \right| < \frac{\epsilon}{300}.$$

*Further, events in Lemma 24 hold.*

*Proof.* We will first calculate the projection of the gradient on $e_{S[i'],b'}$. When the event in Lemma 23 happens,

$$\left\langle \frac{d\mathcal{L}^{(2)}}{dW_{r,d+1:2d}}, e_{S[i'],b'} \right\rangle = h_r \sum_{i=n+1}^{n+k} \sum_{b_1=0}^{1} \sum_{b_2=0}^{1} \mathbf{1}(\nu_{r,i,b_2} > 0)(-1)^{b_1+b_2} \langle e_{S[i'],b'}, e_{S[i],b_1} \rangle + O(\frac{1}{n^8}).$$

By Lemmas 5 and 30, with probability $1 - 0.01\delta$,

$$\left| \left\langle \frac{d\mathcal{L}^{(2)}}{dW_{r,d+1:2d}}, e_{S[i'],b'} \right\rangle - \text{sign}(\nu_{r,i',b'}) h_r \mathbf{1}(\nu_{r,i',0}\nu_{r,i',1} < 0) \right| \leq \frac{1}{200m}$$

By Lemmas 5 and 17, we have that,

$$\left| \left\langle W_{r,d+1:2d}^{(2)}, e_{S[i'],b'} \right\rangle + \text{sign}(\nu_{r,i',b'}) h_r \eta_2 \mathbf{1}(\nu_{r,i',0}\nu_{r,i',1} < 0) \right| < \frac{(\eta_0 + \eta_1)}{20mn} + \frac{\eta_2}{200m} < \frac{\eta_2}{100m}.$$

Combining with Lemma 24 and $\eta_2 = \frac{4\epsilon m}{3}$, we have the result. $\qquad\square$

**Lemma 26.** *With probability $1 - \delta$, for all $\boldsymbol{b} \in \{0,1\}^{n+k}$, $i \in [n+1, n+k]$, we have that*

$$\left| \mathcal{T}(\mathbf{E}(\boldsymbol{b}))[i] - \frac{\epsilon(-1)^{\boldsymbol{b}[i+1]+1}}{3} \right| < \frac{\epsilon}{3}.$$

*Proof.* By Lemma 23, the following holds,

$$\left\langle W_{r,1:d}^{(3)}, \mathbf{E}(\mathbf{b})[i] \right\rangle \mathrm{sign}(\nu_{r,i,\mathbf{b}[i]}) \in [5\epsilon/6, 7\epsilon/6].$$

Combining with Lemma 25,

$$\left| \left\langle W_{r,d+1:2d}^{(2)}, \mathcal{A}(A^3)[\mathbf{E}(\mathbf{b})][i] \right\rangle + \mathrm{sign}(\nu_{r,i,\mathbf{b}[S[i]]} h_r) \frac{2\epsilon}{3} \mathbf{1}(\nu_{r,i,0}\nu_{r,i,1} < 0) \right| < \frac{\epsilon}{300}.$$

Hence, we still have that

$$| \left\langle W_{r,d+1:2d}^{(3)}, \mathcal{A}(A^3)[\mathbf{E}(\mathbf{b})][i] \right\rangle | < \frac{5\epsilon}{6}.$$

which implies that,

$$\mathbf{1}(\left\langle W_{r,1:d}^{(3)}, \mathbf{E}(\mathbf{b})[i] \right\rangle > 0) = \mathbf{1}(\nu_{r,i,\mathbf{b}[i]} > 0).$$

Consider the output contribution of the attention part,

$$\left| \sum_{r=1}^{2m} h_r \left\langle W_{r,d+1:2d}^{(3)}, \mathcal{A}(A^3)[\mathbf{E}(\mathbf{b})][i] \right\rangle \mathbf{1}(\nu_{r,i,\mathbf{b}[i]} > 0) \right.$$
$$\left. + \sum_{r=1}^{2m} \frac{2h_r\epsilon}{3} \mathrm{sign}(\nu_{r,i,\mathbf{b}[S[i]]} h_r) \mathbf{1}(\nu_{r,i,0}\nu_{r,i,1} < 0, \nu_{r,i,\mathbf{b}[i]} > 0) \right| < \frac{\epsilon}{300}.$$

The later term with $1 - \delta/100$ satisfies,

$$\sum_{r=1}^{2m} \frac{2h_r\epsilon}{3} \mathrm{sign}(\nu_{r,i,\mathbf{b}[S[i]]} h_r) \mathbf{1}(\nu_{r,i,0}\nu_{r,i,1} < 0, \nu_{r,i,\mathbf{b}[i]} > 0)$$
$$= \sum_{r=1}^{2m} \frac{\epsilon}{3m} \mathrm{sign}(\nu_{r,i,\mathbf{b}[S[i]]}) \mathbf{1}(\nu_{r,i,0}\nu_{r,i,1} < 0, \nu_{r,i,\mathbf{b}[i]} > 0)$$
$$= \sum_{r=1}^{2m} \frac{\epsilon}{3m} (-1)^{\mathbf{b}[S[i]]+\mathbf{b}[i]} \mathbf{1}(\nu_{r,i,0}\nu_{r,i,1} < 0)$$
$$= \frac{\epsilon}{3} (-1)^{\mathbf{b}[S[i]]+\mathbf{b}[i]} + O(\frac{\epsilon \log(100n/\delta)}{\sqrt{m}})$$

This shows that,

$$\left| \sum_{r=1}^{2m} h_r \left\langle W_{r,d+1:2d}^{(3)}, \mathcal{A}(A^3)[\mathbf{E}(\mathbf{b})][i] \right\rangle \mathbf{1}(\nu_{r,i,\mathbf{b}[i]} > 0) - \frac{\epsilon}{3}(-1)^{1+\mathbf{b}[S[i]]+\mathbf{b}[i]} \right| \leq \frac{\epsilon}{150}.$$

On the other hand, we have that

$$\left| \sum_{r=1}^{2m} h_r \left\langle W_{r,1:d}^{(3)}, \mathbf{E}(\mathbf{b})[i] \right\rangle \mathbf{1}(\nu_{r,i,\mathbf{b}[i]} > 0) \right|$$
$$\leq \left| \sum_{r=1}^{2m} h_r \left\langle W_{r,1:d}^{(0)}, \mathbf{E}(\mathbf{b})[i] \right\rangle \mathbf{1}(\nu_{r,i,\mathbf{b}[i]} > 0) \right| + \left| \sum_{r=1}^{2m} h_r \left\langle W_{r,1:d}^{(3)} - W_{r,1:d}^{(0)}, \mathbf{E}(\mathbf{b})[i] \right\rangle \mathbf{1}(\nu_{r,i,\mathbf{b}[i]} > 0) \right|$$
$$\leq \frac{\epsilon}{4}.$$

The first term is bounded due to standard concentration inequality over the axis of $r$. The second term is bounded by Lemmas 17 and 23. Combining the terms, we have that

$$\left| \mathcal{T}(\mathbf{E}(\mathbf{b}))[i] - \frac{\epsilon(-1)^{\mathbf{b}[i]+\mathbf{b}[S[i]]+1}}{3} \right| = \left| \mathcal{T}(\mathbf{E}(\mathbf{b}))[i] - \frac{\epsilon(-1)^{\mathbf{b}[i+1]+1}}{3} \right| < \frac{\epsilon}{3}.$$

This concludes that the model is able to predict the correct output. $\qquad\square$

### B.5 FINAL PROOF

**Lemma 27.** *The results in Lemmas 18 and 26 can be extended to hinge loss $\ell(\hat{y}, y) = \max\{(-1)^y \hat{y} + 1, 0\}$ with the same probability.*

*Proof.* Our Lemmas 17 and 23 shows that the output of the FFN layer is bounded by $\epsilon/3$ throughout training. The hinge loss is linear when the output is in $[-1, 1]$. Hence, the results can be directly extended to hinge loss. $\qquad\square$

*Proof of Theorem 3.* The original theorem is a combination of Lemmas 18, 26 and 27. $\qquad\square$

### B.6 TECHNICAL LEMMA

**Lemma 28** (Theorem 4.4.5 of Vershynin (2018))**.** *There exists universal constant $C$, let $A$ be an $m \times n$ random matrix whose entries $A_{ij}$ are independent, mean zero, sub-Gaussian random variables. Then, for any $t > 0$, we have*
$$\|A\| \leq CK\left(\sqrt{m} + \sqrt{n} + t\right)$$
*with probability at least $1 - 2\exp(-t^2)$. Here $K = \max_{i,j} \|A_{ij}\|_{\psi_2}$ is the maximum sub-Gaussian norm of $A_{ij}$.*

**Lemma 29** (Theorem 4.6.1 of Vershynin (2018))**.** *Let $A$ be an $m \times n$ matrix whose rows $A_i$ are independent, mean zero, sub-Gaussian isotropic random vectors in $\mathbb{R}^n$. Then, for any $t \geq 0$, we have*
$$\sqrt{m} - CK^2(\sqrt{n} + t) \leq s_n(A) \leq s_1(A) \leq \sqrt{m} + CK^2(\sqrt{n} + t)$$
*with probability at least $1 - 2\exp(-t^2)$. Here, $K = \max_i \|A_i\|_{\psi_2}$.*

*Furthermore, a slightly stronger conclusion holds:*
$$\left\| \frac{1}{m} A^\top A - I_n \right\| \leq K^2 \max(\delta, \delta^2),$$
*where $\delta = C\left(\sqrt{\frac{n}{m}} + \frac{t}{\sqrt{m}}\right)$.*

**Lemma 30.** *Define $v = \sum_{i=n+1}^{n+k} \sum_{b=0}^{1} \lambda_{i,b} e_{i,b}$, then with probability $1 - \delta$,*

$$|\langle e_{i',b'}, v\rangle - \lambda_{i',b'}| \leq \sqrt{\frac{2\log\left(\frac{2}{\delta}\right)}{d} \sum_{(i,b) \neq (i',b')} \lambda_{i,b}^2}.$$

*Proof.* We aim to bound $|\langle e_{i',b'}, v\rangle|$, where

$$v = \sum_{i=n+1}^{n+k} \sum_{b=0}^{1} \lambda_{i,b} e_{i,b}.$$

Note that each $e_{i,b} \in \mathbb{R}^d$ has entries that are independent random variables from $\left\{-\frac{1}{\sqrt{d}}, \frac{1}{\sqrt{d}}\right\}$.

First, observe that

$$\langle e_{i',b'}, v\rangle = \sum_{(i,b)} \lambda_{i,b} \langle e_{i',b'}, e_{i,b}\rangle = \lambda_{i',b'} + \sum_{(i,b) \neq (i',b')} \lambda_{i,b} \langle e_{i',b'}, e_{i,b}\rangle.$$

Since $\langle e_{i',b'}, e_{i',b'} \rangle = 1$ and for $(i,b) \neq (i',b')$, the inner products $\langle e_{i',b'}, e_{i,b} \rangle$ are sums of independent random variables with mean zero.

Define
$$X_{i,b} = \lambda_{i,b} \langle e_{i',b'}, e_{i,b} \rangle, \quad \text{for } (i,b) \neq (i',b').$$

Each $X_{i,b}$ is a sum of $d$ independent random variables bounded in $\left[ -\frac{|\lambda_{i,b}|}{d}, \frac{|\lambda_{i,b}|}{d} \right]$, because each component $e_{i',b',j} e_{i,b,j}$ is $\pm \frac{1}{d}$.

By the Azuma-Hoeffding inequality, for any $t > 0$,

$$\Pr\left( \left| \sum_{(i,b) \neq (i',b')} X_{i,b} \right| \geq t \right) \leq 2 \exp\left( -\frac{2dt^2}{\sum_{(i,b) \neq (i',b')} 4\lambda_{i,b}^2} \right) = 2 \exp\left( -\frac{dt^2}{2 \sum_{(i,b) \neq (i',b')} \lambda_{i,b}^2} \right).$$

Setting the right-hand side equal to $\delta$ and solving for $t$, we get

$$t = \sqrt{ \frac{2 \log\left( \frac{2}{\delta} \right)}{d} \sum_{(i,b) \neq (i',b')} \lambda_{i,b}^2 }.$$

Therefore, with probability at least $1 - \delta$,

$$|\langle e_{i',b'}, v \rangle - \lambda_{i',b'}| \leq \sqrt{ \frac{2 \log\left( \frac{2}{\delta} \right)}{d} \sum_{(i,b) \neq (i',b')} \lambda_{i,b}^2 }.$$

The proof is then complete. $\qquad \square$

**Lemma 31.** *With probability $1 - \delta$, it holds that*

$$\forall i, i' \in [n+k], b, b' \in \{0,1\}, \| e_{i,b}^T e_{i',b'} - \mathbf{1}(i = i' \& b = b') \|_2 \leq \sqrt{ 2 \frac{\log(8nd/\delta)}{d} }.$$

*Proof.* This is a direct consequence combining Lemma 30 and union bound. $\qquad \square$

**Lemma 32.** *If for constant $C$, the attention score before softmax has the following property:*

- *for each position $i$, the target position $j = S[i]$ satisfies:*

$$\left\langle e_{j,b_1'}, A^{(2)} e_{i,b_2'} \right\rangle \geq 2C \log n.$$

- *For all other positions $j' \neq j$, the attention weights satisfy:*

$$\left| \left\langle e_{j',b_1'}, A^{(2)} e_{i,b_2'} \right\rangle \right| \leq C \log n.$$

*The attention output is approximately one-hot, with*

$$\left| \mathcal{A}[A^2](\mathbf{E}(\boldsymbol{b}))[i] - e_{S[i],\boldsymbol{b}[S[i]]} \right| < 4/n^{C-1}.$$

*Proof of Lemma 22.* The attention output for position $i$ is given by:

$$\mathcal{A}[A^2](\mathbf{E}(\mathbf{b}))[i] = \mathrm{X} \cdot \mathrm{softmax}\left( \mathrm{X}^\top A \mathrm{X} \right)[i],$$

where the softmax is applied column-wise.

Given the condition from Lemma 5, we set

$$\Delta = 2C \log n$$

This implies that:

$$e^{-\Delta/2} < e^{-C \log n} = \frac{1}{n^C}.$$

Define $z_{j'} = \langle e_{j',b_1'}, A^{(2)} e_{i,b_2'} \rangle$.

The softmax for the target position $j = S[i]$ is:

$$\text{softmax}(z)_j = \frac{e^{z_j}}{e^{z_j} + \sum_{j' \neq j} e^{z_{j'}}}.$$

Given that $z_j \geq \Delta$ and $|z_{j'}| \leq \frac{\Delta}{2}$ (since $\frac{\eta_0 \eta_1}{512mn^2} = \frac{\Delta}{2}$), we have:

$$z_j - z_{j'} \geq \Delta - \frac{\Delta}{2} = \frac{\Delta}{2}.$$

Thus:

$$\text{softmax}(z)_j \geq \frac{e^{\Delta}}{e^{\Delta} + (T-1)e^{\Delta/2}} = \frac{1}{1 + (T-1)e^{-\Delta/2}}.$$

Therefore with $T \leq 2n$:

$$\text{softmax}(z)_j \geq \frac{1}{1 + (T-1) \cdot \frac{1}{n^C}} \geq 1 - \frac{2}{n^{C-1}}.$$

Therefore

$$\sum_{j' \neq j} \text{softmax}(z)_{j'} \leq \frac{2}{n^{C-1}}.$$

The attention output for position $i$ is:

$$\mathcal{A}[A^2](\mathbf{E}(\mathbf{b}))[i] = \sum_j e_{j,\mathbf{b}[j]} \text{softmax}(z)_j.$$

Substituting the bounds:

$$\left| \mathcal{A}[A^2](\mathbf{E}(\mathbf{b}))[i] - e_{S[i],\mathbf{b}[S[i]]} \right| = \left| \sum_{j' \neq S[i]} e_{j',\mathbf{b}[j']} \text{softmax}(z)_{j'} \right| + \left| e_{S[i],\mathbf{b}[S[i]]} (\text{softmax}(z)_{S[i]} - 1) \right|$$

$$\leq \sum_{j' \neq S[i]} \text{softmax}(z)_{j'} + \left| (\text{softmax}(z)_{S[i]} - 1) \right| \leq \frac{4}{n^{C-1}}$$

We conclude our proof. $\qquad\qquad\qquad\qquad\qquad\qquad\qquad\qquad\qquad\qquad\qquad\qquad\qquad\square$

## C  ADDITIONAL EXPERIMENT RESULTS

In this section, we provide details of the experiment setup and present additional results. All training was conducted using PyTorch Paszke et al. (2019) on NVIDIA RTX A10 GPUs.

### C.1  EXPERIMENT DETAILS

The transformer architecture adopted in the experiment section is based on the GPT-2 model (Radford et al., 2019) with a hidden size of 720, an intermediate size of 3072, and trainable position embeddings. For all experiments, we use `Adam` (Kingma, 2014) optimizer with random initialization, using hyperparameters $\beta_1 = 0.9, \beta_2 = 0.999$, a weight decay of 0 and a linear decay learning rate schedule. The batch size is set to 512 throughout. The validation set contains 2048 samples which are nonintersecting with the training data. In all experiments regarding sample complexity, a test set of size 2048 which is non-intersected with the training data is used.

**Details of the experiments in Section 4.3**  This section examines the normalized attention entropy of Qwen2-7B (Yang et al., 2024) and Qwen2-Math-7B models (Qwen, 2024) on the GSM8K dataset (Cobbe et al., 2021) with and without CoT prompting respectively. Both models consist of 28 layers, each with 28 attention heads. The normalized attention entropy is computed as the average across the GSM8K test set. In Figure 6, the entropy values of each attention head in a layer are sorted separately under the three setups. While Figure 6 presents the normalized attention entropy for specific attention heads in the first, last, and two intermediate layers, Figure 7 shows the entropy across all layers.

For `With CoT` data, the input is concatenated with the ground truth answer from the GSM8K dataset. For `With CoT` data, we extract the final answer from the ground truth, and concatenate the input with the string "The answer is [Final Answer].".

### C.2  ADDITIONAL RESULTS

In Figure 3 (Right), we present the attention pattern of a single-layer, single-head transformer trained on the $(n = 20, k = 40)$ parity problem with CoT data. In Figure 8, we show the attention patterns of a multi-layer, multi-head transformer trained on the same problem. We observe that in the first layer, the attention pattern is sparse and interpretable, with each secret variable attended to by at least one attention head. In contrast, the second layer exhibits an almost uniform attention pattern. A possible explanation is that the first layer captures sufficient information, which is then transferred to subsequent layers via the residual connections.

In Section 4.2, we observe that training with repeated data can help Transformers learn the parity function, but it still requires significantly more computation compared to trained on CoT data (Figure 4). To further substantiate this observation, we explore the training of models with and without CoT on the $(n = 20, k = 12)$ parity problem. Compared with the $(n = 20, k = 6)$ parity problem considered in Figure 4, this problem is harder for the models to learn as the number of secret variables $k$ is larger. We examine various configurations: the number of layers and heads ranges from $1, 2, 3, 4, 6,$ and $8$; the learning rate varies from $6 \times 10^{-6}$, $8 \times 10^{-8}$, to $1 \times 10^{-4}$; and the training dataset size varies from $10^4$, $10^5$ to $10^6$, with corresponding epochs ranging from 1000, 100 to 10. Across all configurations we examined, training without CoT fails to achieve non-trivial accuracy (Figure 9). In contrast, models trained with CoT achieve perfect evaluation accuracy when trained on a dataset with $10,000$ samples for only 6 epochs, or with approximately $60,000$ fresh samples in one-pass training setting.

In Figure 5, a 4-layer 4-head transformer achieves perfect evaluation accuracy on $(n = 20, k = 6)$ problem when trained on a small dataset of 50000 samples, but fails to achieve non-trivial accuracy when trained on a larger dataset. Furthermore, successful learning coincides with a significant decrease in attention entropy, indicating the development of sparse attention, while entropy remains high when trained on a larger dataset. In Figure 10, we present more results across different architectures (1-layer 1-head, 2-layer 3-head and 4-layer 4-head transformers) with dataset size of $5000, 10000, 50000, 100000, 1000000$, and observe the same pattern.

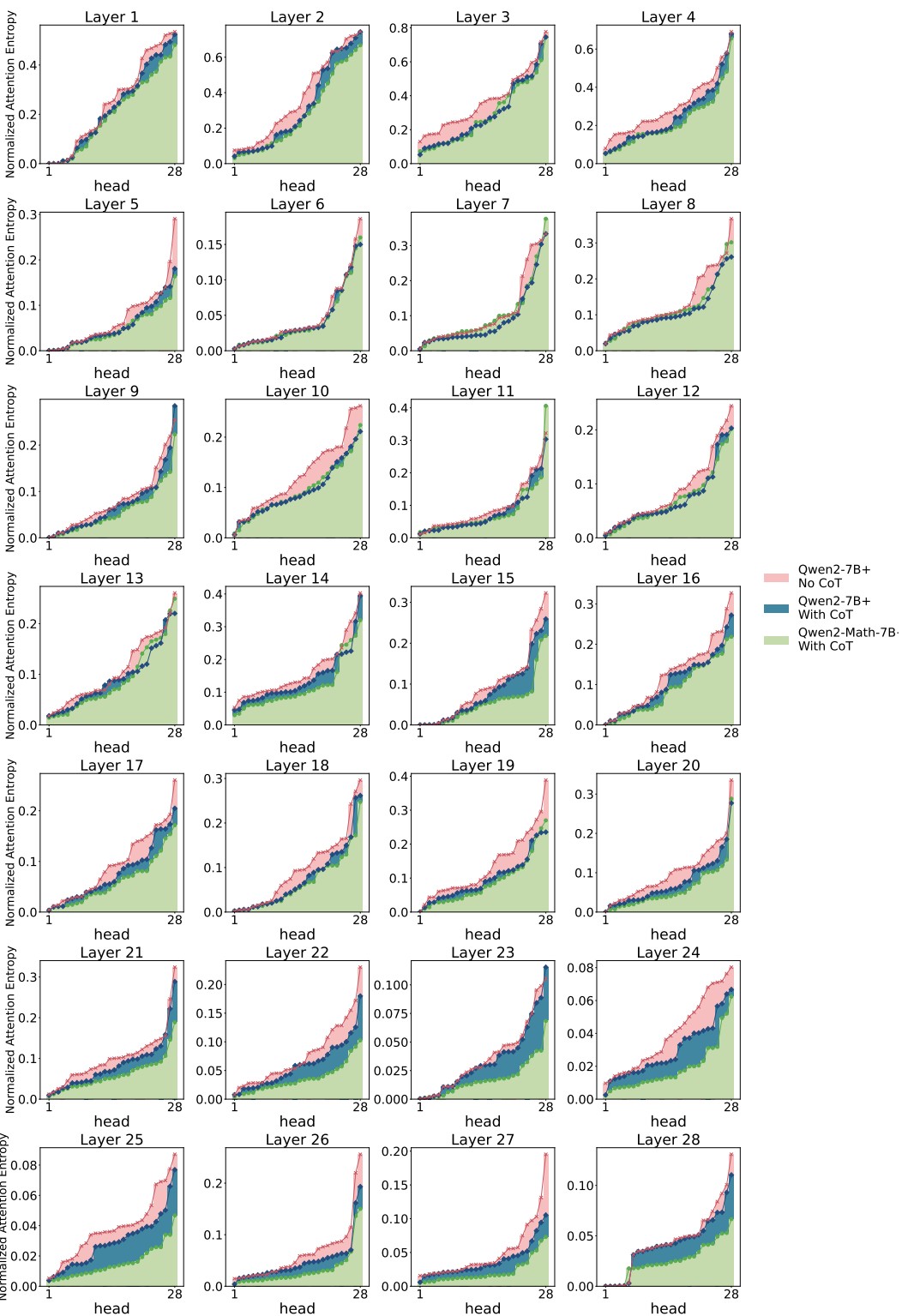

Figure 7: The normalized attention entropy of the pre-trained Qwen2-7B and math-specialized Qwen2-Math-7B models on the GSM8K dataset with and without CoT prompting (Section 4.3). Each bar represents the entropy of an attention head.

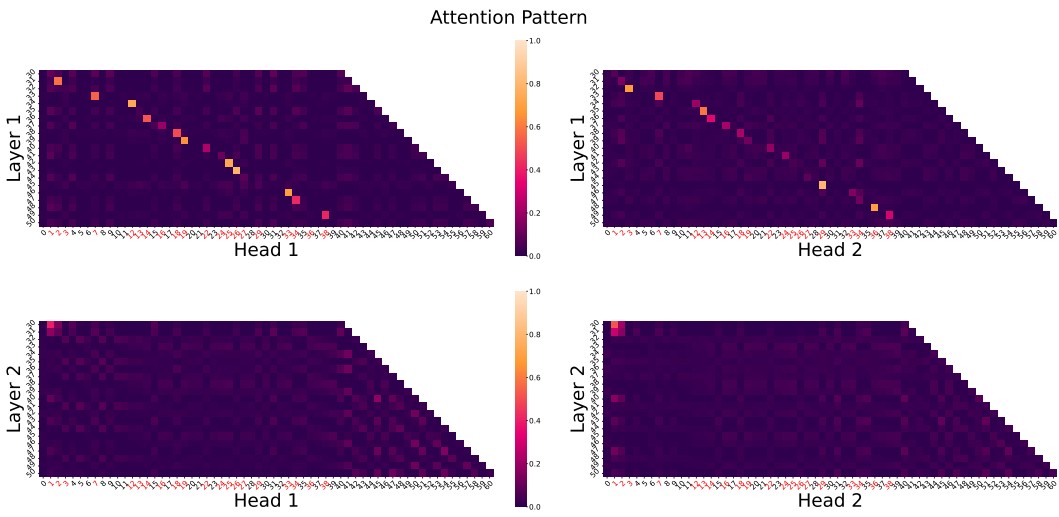

Figure 8: The attention pattern learned by a 2-layer 2-head transformer on $(n = 20, k = 6)$ parity problem with CoT.

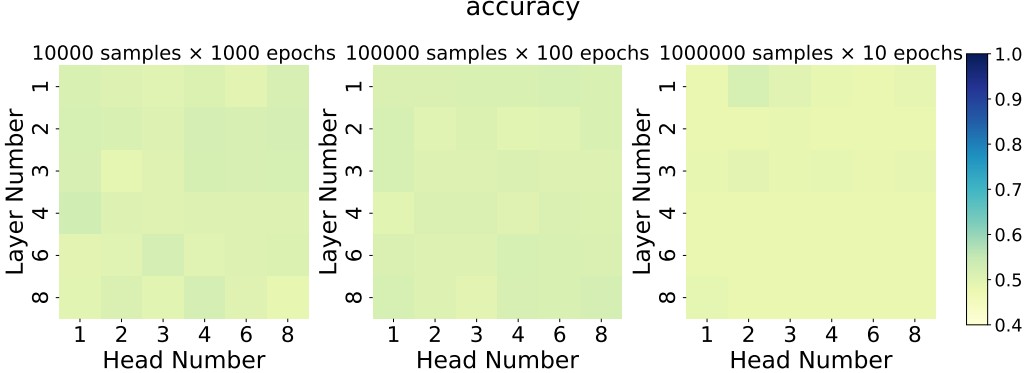

Figure 9: Training with CoT on $(n = 20, k = 12)$ parity problem fails to achieve non-trivial accuracy under different configurations.

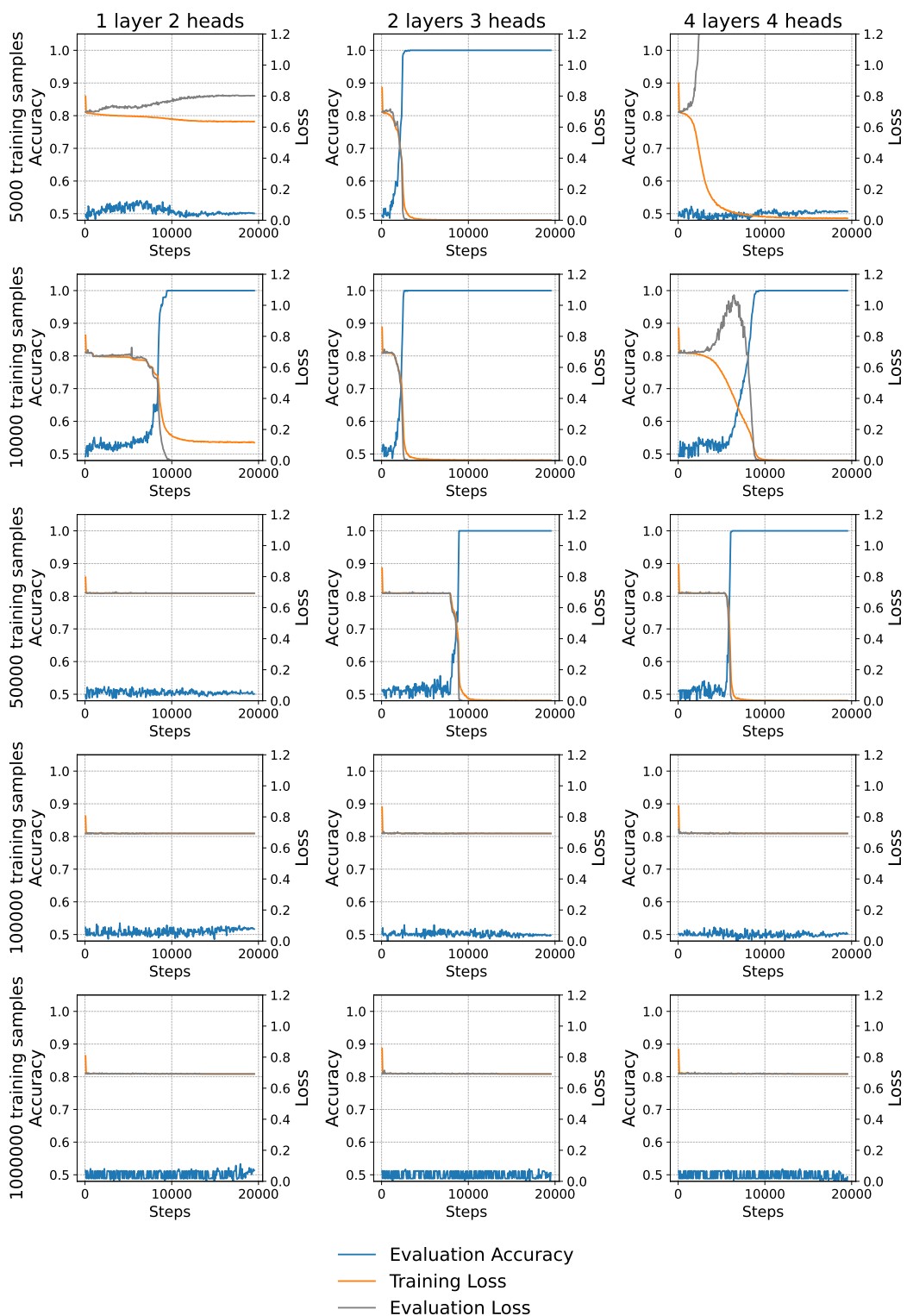

Figure 10: Transformer trained on the $(n = 20, k = 6)$ parity problem without CoT with different sizes of training dataset and a fixed number of iterations.

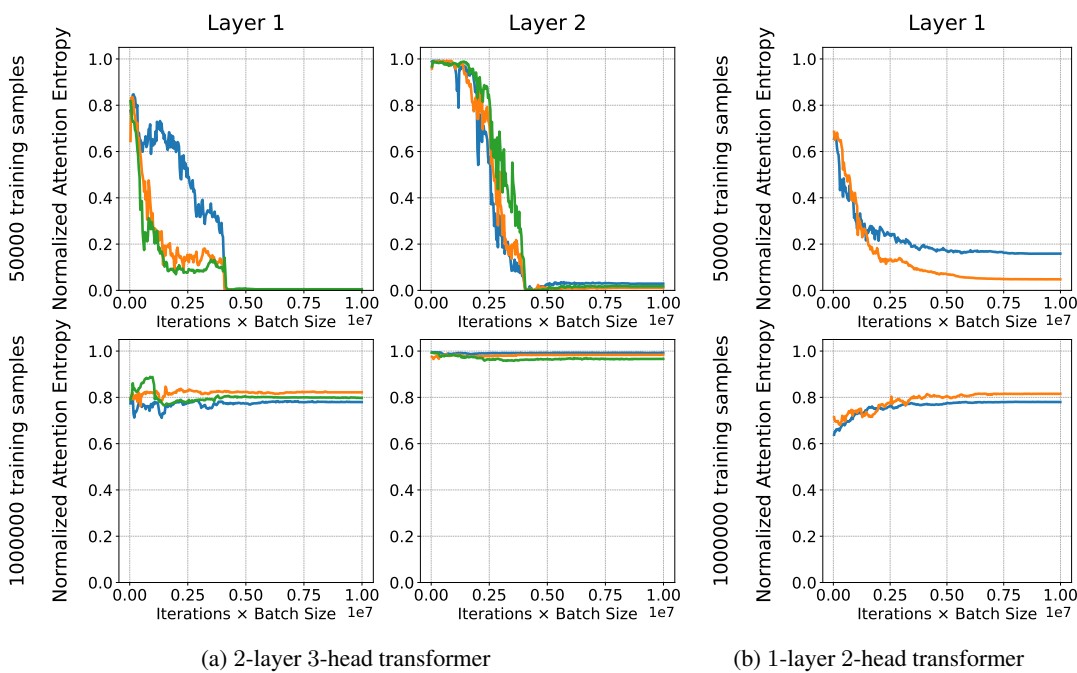

(a) 2-layer 3-head transformer

(b) 1-layer 2-head transformer

Figure 11: Normalized attention entropy curve of transformers training on the $(n = 20, k = 6)$ parity problem without CoT. Each line in the graph represents the attention entropy for a head of a certain layer.

