# OpenReview forum: "From Sparse Dependence to Sparse Attention: Unveiling How Chain-of-Thought Enhances Transformer Sample Efficiency"
_ICLR.cc/2025/Conference — ICLR 2025 Poster_

### Official Review · Reviewer_ezDX · 2024-10-28

**Soundness:** 3
**Presentation:** 3
**Contribution:** 4
**Rating:** 6
**Confidence:** 4

**Summary:**

This paper presents a theory to explain why Chain-of-Thought (CoT) helps Transformer models to learn complex tasks. The paper first shows that a Transformer architecture with one layer and one head is expressive enough to represent the parity learning task. However, exponentially many samples are needed to solve this task with a subquadratic memory. Then, the authors demonstrate that a reformulation of the task with CoT can be solved with only a linear number of samples.

**Strengths:**

This paper studies the learning dynamics of a Transformer layer (with non-linear attention and MLP sublayers), which is crucial in order to understand better the properties of this architecture, and a technically difficult problem due to the highly non-linear parameterization of the architecture. The analysis of CoT is original and compelling, in that it gives a provable mechanism by which CoT simplifies the learning problem.

**Weaknesses:**

While the ideas in the paper are nice, the paper would highly benefit from polishing to improve the clarity and presentation of the message. In particular, the proofs are written in a telegraphic style, which makes it particularly difficult to check correctness. It also makes it hard to get a sense of how general the approach is, where the specific initialization scheme comes into play, etc. I give below more specific examples of things that confused me and would benefit from cleaning up.

Besides, I am not convinced by the real-world experiments. The difference in terms of entropy of the attention layers does not seem very significant, so in my opinion it is difficult to reach any strong conclusion from the figures.

All in all, the rating reflects the fact that the ideas are interesting, but I believe polishing the paper would make it much better.

Examples of confusing statements in the main text and the beginning of the proofs (the only point for which I would like an answer during the rebuttal is the first one, others are less important):
- in Theorem 2, the condition on $d$, $m$, $n$ and $k$ is not satisfied for the dimensions taken in the proof of Theorem 1, so it is hard to relate both results.
- line 296: “which only applies to one-pass training”, but Theorem 2 applies to multipass training?
- line 831: why $m>100$?
- line 841: I agree that the results should follow from random matrix theory, but you should give a proper statement and proof/reference. The result depends on the relative value of $d$ and $k$, and the current statement $d = \tilde{\Omega}(k)$ is too hand-wavy to conclude that the specific numerical values given in the Lemma are verified.
- line 844: why $\delta/4$?
- lines 896: why $8$ instead of $10$?
- line 901: the $\mathcal{O}(1/n^7)$ needs to be explained.
- line 913: “by choosing $a_r$ and $b_r$ iteratively”: this needs to be explained.
- line 916: “the parameters can be encoded in $\mathcal{O}(\log n)$ precision”. Why?
- line 960: this is not a valid proof of the Lemma, the derivation of the inequalities is non-trivial from the Assumption.

Minor remarks (no impact on score):
- line 46: the specific example of the number of letters in a word is not very relevant, because perhaps this is precisely an example of a task which may not be representable by a Transformer model due to the tokenisation mechanism.
- line 77: $k$ is not introduced yet.
- line 191: the bounds of the sum should depend whether it is a CoT or non-CoT task.
- line 230: it should be $W_{r,1:m}$ instead of $W_{r,1:d}$.

**Questions:**

In Definition 3, is there any advantage to taking random embeddings vectors, instead of fixed orthogonal vectors? From a modelling perspective, I don’t think random embeddings add much, and taking them fixed would greatly simplify the proofs by cancelling many terms, which you currently have to bound using random matrix theory and concentration inequalities. This contributes to making the proofs difficult to follow.

---

> ### Author Response · Authors · 2024-11-24
>
> We thank the reviewer for acknowledging this paper as having a nice and original idea. We especially appreciate the reviewer’s detailed suggestions.  Following the reviewer’s suggestion, we have polished our writing and proofs.
>
> Before delving into the detailed edits, we would like to point out that our proof of Theorem 3 is technically challenging to simplify because we are analyzing **finite-batch** dynamics with **random embedding**. While our proof for Theorem 3 can be greatly simplified without these two constraints, both constraints are necessary for us to derive a sample complexity upper bound under the regime of Theorem 2\.
>
> To improve the readability of our proof of Theorem 3, we now format the proof to first define and bound necessary auxiliary statistics in Section A.4.1. We also include a table listing out the definitions and orders of these auxiliary statistics. We then analyze the trajectory of weight step by step in Section A.4.2 to A.4.4, where each section corresponds to a different step. Finally, we conclude our proof in Section A.4.5 and prove our technical lemmas in Section A.4.5.
>
> We would like to address the reviewers’ concerns below. We will start by addressing high-level concerns and list detailed edits to the end. We are happy to make further edits.
>
> **W1.1: Unclear Role of Initialization Scheme**
>
> The initialization we used is mainly used to guarantee the activation pattern of the MLP (i.e. whether the $r$-th neuron is activated by the ReLU activation) is solely determined by the position $i$, the boolean value $\\mathbf b\[i\]$ and the randomness of $v\_{r,i,\\mathbf b\[i\]}$ since $\\langle e\_{i,\\mathbf b\[i\]},W\_{r,1:d}\\rangle \\approx v\_{r,i,b}$ with high probability. As a result, for a fixed $(i,\\mathbf b\[i\])$, the FFN can be viewed as a linear function, which consists of the neurons with $v\_{r,i,\\mathbf b\[i\]}\>0$. At phase 1, this linear function will develop a stronger correlation with embedding $e\_{(S\[i\],\\mathbf b\[j\])}$. We have updated our manuscript to state this clearly in the main text.
>
> **W1.2: Confusing details of Theorem 1**
>
> In our theoretical analysis, we assume that $k \\in \[n / \\log^5 (n / \\delta), n / \\log^4(n/\\delta)\]$. Hence the construction in Theorem 1 will fall in the scope of Theorem 2\. We thank the reviewer for pointing out this missing assumption and have added it in the updated version.
>
> **W2: The real-world experiments are not convincing enough.**
>
> For the real-world experiments, we would like to clarify and highlight the following points in response:  As shown in **Figure 7** in the appendix, the attention entropy patterns across all layers of the Transformer model **consistently** reflect the trend discussed in the paper: 1\. The attention entropy is lower when processing With CoT data compared to No CoT data (**18.0%** reduction on average across all layers), indicating that real-world CoT data indeed exhibits a sparser structure; 2\. On the same With CoT data, Qwen2-Math-7B model demonstrates lower attention entropy compared to Qwen2-7B model (**17.4%** reduction on average across all layers), suggesting that fine-tuning on CoT data promotes the development of sparser attention patterns. We argue that these reductions (18.0% and 17.4%) are **statistically significant,**  indicating the sparse structure inherent in CoT data as well as the development of sparser attention from training on CoT data. )
>
> **Q1: Why do we use random embedding instead of orthogonal ones?**
>
> We agree that using random embedding adds complexity to our analysis. However, this complexity is necessary because when we use orthogonal embeddings, the number of parameters in the model will not fall into the regime of Theorem 2 and our lower bound will no longer hold. Hence, if we use orthogonal embeddings, we could not show theoretically there is an exponential sample complexity gap in this case.

---

> > ### Author Response · Authors · 2024-11-24
> >
> > **Detailed Edits**
> >
> > **W1.3: In line 296, we stated that our theory “only applies to one-pass training”**
> >
> > This is a typo; our theory only applies to constant-pass training.
> >
> > **W1.4: In line 831, we assume “m \> 100”**
> >
> > This is only a technical assumption to ensure that we have a large enough $m$ to ensure the constants in our bounds are valid.
> >
> > **W1.5: In line 841, we omit the proof for Lemma 1\.**
> >
> > We have updated our draft to include a more detailed proof for Lemma 1\.
> >
> > **W1.6: Why do we use $\\delta / 4$ in some of our lemmas?**
> >
> > We intentionally divide $\\delta$ by some constant to apply a union bound over event in the end to get the exact probability bound $\\delta$.
> >
> > **W1.7: Why 8 instead of 10 on line 896**
> >
> > This is a typo and we mean $\> 8 \\log n$. The key here is that we have a large enough difference (\\Theta(\\log n)) in the attention score to get one-hot attention and the exact constant is not essential to the proof.
> >
> > **W1.8: Why $1/n^7$ on line 971**
> >
> > Given the difference in attention score, we could bound how close the attention is to a one-hot attention. We have added the proof for this inequality.
> >
> > **W1.9: We should concretely state how $a\_r$ and $b\_r$ are chosen.**
> >
> > We thank the reviewer for the valuable suggestions and have now written an analytical form of $a\_r$ and $b\_r$ in the construction. We have also improved our construction so that $a\_r$ and $b\_r$ are now bounded by a polynomial of $n$.
> >
> > **W1.10: Why parameters can be encoded in $O(\\log n)$ precision?**
> >
> > This is because all the parameters and the width of the constructions are all upper bounded by polynomials of $n$ and the depth of the construction is 1\. We can then control the approximation error with $O(\\log n)$ precision. We have added a discussion in the current draft.
> >
> > **W1.11: We should detail our proof of Lemma 5\.**
> >
> > We thank the reviewer for the suggestion and have detailed the derivation.
> >
> > **W1.12: The example of counting letters in the word is not highly relevant.**
> >
> > We believe counting letters in the word is representable by a small Transformer because if Transformers memorize the letters for each token, then counting letters reduces to an addition problem, which is representable.
> >
> > **W1.13: We use $k$ in our introduction without definition.**
> >
> > We have removed the dependence of $k$ in our introduction.
> >
> > **W1.14: The definition of loss function in the main text only holds for CoT data.**
> >
> > Sorry for the typo. We have corrected it in the current draft.
> >
> > **W1.15: Typo on line 230\.**
> >
> > Sorry for the typo. We have corrected it in the current draft.

---

> > > ### Comment · Reviewer_ezDX · 2024-11-26
> > >
> > > Thank you for your detailed answer. I appreciate that the authors have made a very substantial effort to improve the readability of their proofs. I increased my score.

---

### Official Review · Reviewer_3yRE · 2024-10-31

**Soundness:** 2
**Presentation:** 4
**Contribution:** 3
**Rating:** 6
**Confidence:** 4

**Summary:**

This paper theoretically studies the training dynamics of one-layer Transformers learning sparse parity problems using zero-shot CoT. The authors especially compare the sample complexity with or without CoT, which is interesting and verified by the experiments. The experiments and discussion on multi-pass training are also interesting.

**Strengths:**

The paper is well-written and easy to follow.
The experiments are overall solid to support the theory.
It is novel to study parity problems using Transformers. The theoretical analysis is novel. The comparison between CoT and without CoT is quite impressive and significant in future research.

**Weaknesses:**

There are two major issues.
1. The theoretical proof seems incomplete. I failed to find proof of Theorem 3. I can only find a proof sketch and some supportive lemmas. As a theoretical paper, this harms the soundness.
2. If my understanding is correct, during the training and evaluation, the location to compute the parity in every sequence is the same, e.g., we always use the location 1,2, and 5 to compute the parity. I think it is a stronger and more reasonable result to make the location not fixed. For example, let $k=3$, then I can learn the sparse parity with any three locations. Especially during the testing, I can compute parity on a certain tuple of three locations that never appear during the training. This can make your work a stronger CoT inference rather than more of a training analysis.

**Questions:**

1. Can your analysis be extended to few-shot CoT?
2. In lemma 22, line 1834, is $W^{(3)}$ in the third phase of the training? If so, then there is only one step in each phase? Does it imply that the training process is too fast or simple? This seems unrealistic.
3. This work seems to remove positional encoding. Why? If I add positional encoding, can the sample complexity be reduced in both cases, i.e., with or without CoT?

---

> ### Author Response · Authors · 2024-11-24
>
> We thank the reviewer for acknowledging the paper's novelty and clarity and address their questions below.
>
> **Why do we study the fixed secret parity setup?**
>
> In Weakness 2, the reviewer asked whether the location of the secret is fixed during training and inference time. This is indeed correct. We would like to restate the motivation for our study on the parity problem with a fixed secret key.
>
> 1. We aim to study how Chain of Thought (CoT) improves sample complexity in reasoning.
>    1. Hence, we choose learning a binary secret serves as a **minimal** example of learning an algorithm
>    2. We both lower bound the sample complexity without CoT and upper bound the sample complexity with CoT, showing that there is an exponential sample complexity gap.
>    3. Given our lower bound, we believe that the simplicity of the fixed-secret learning task is indeed a strength rather than a limitation. Our lower bound shows that without CoT, Transformers will struggle to learn even simple algorithms like parity, let alone other more complex algorithms.
> 2. We intentionally omitted a few-shot examples to focus on studying the role of CoT independently.
>    1. It has been shown empirically that models can learn to perform CoT without any prompting or in-context examples.
>    2. As our primary goal is to understand the role of CoT instead of in-context examples, we deliberately choose not to include few-shot examples in our analysis.
>    3. This is also consistent with previous works \[1,2\] which study CoT in the setting without in-context examples from a representation theory perspective.
> 3. Our results already shed insight into more realistic setups such as real-world math problems.
>    1. While we studied CoT in the synthetic parity task, our results showing that training on CoT data leads to sparser attention is validated on real-world math datasets.
>
>
> \[1\] Towards Revealing the Mystery behind Chain of Thought: A Theoretical Perspective
> \[2\] Chain of Thought Empowers Transformers to Solve Inherently Serial Problems
>
> **W1: It is difficult to find proof for Theorem 3**
>
> In the submitted version, we first outline our proof between line 961 and line 971 and then prove the supporting lemmas for the proof of Theorem 3 in the rest of the appendix. Following the reviewer’s suggestion, we updated our draft and added an explicit pointer to the proof in section A.5.
>
> **W2: We use fixed secret for training and inference**
>
> The location being fixed during training and inference is indeed correct and we have restated our motivation in the response above. We agree with the reviewer that extending this work to a meta-learning setting—where the model would need to learn an algorithm for the parity problem when all queries and keys are provided in the context—would be an exciting direction. However, this presents significantly greater challenges for theoretical analysis, so we leave it for future exploration.

---

> > ### Author Response · Authors · 2024-11-24
> >
> > **Q1: Can we generalize the analysis to learning with a few-shot CoT?**
> >
> > We deliberately choose to analyze CoT without few-shot prompting to focus on the role of CoT as argued above.
> > Meanwhile, our lower bound for sample complexity (Theorem 1\) and representation theory result (Theorem 2\) can be readily extended to a few-shot CoT when a constant number of examples are presented. In terms of the optimization result (Theorem 3), the exact analysis could not directly apply but similar results may be reached through finer analysis. It would also be interesting to analyze whether few-shot examples can reduce sample complexity in future works.
> >
> > **Q2: In our theoretical analysis, training will finish in three steps. Is it too fast and/or unrealistic?**
> > Our analysis indeed shows that the model converges to perfect accuracy within three steps. While this may seem atypical in practical training scenarios, this approach is still meaningful because:
> >
> > 1. **Theoretical Analysis Context:** In prior work on the training dynamics of Transformers and neural networks, models are often analyzed under simplified, constant-step settings. Such simplifications are widely accepted in theoretical studies to isolate and understand fundamental training behaviors. \[1,2\] We would further note that in our theoretical setup, training for more steps with an appropriate learning rate would not hurt the model’s accuracy.
> > 2. **Approximation of Empirical Settings:** In our analysis, we assume a sufficiently large batch size to yield accurate gradient information within each step, enabling rapid convergence. While empirical training often involves smaller batch sizes and noisier gradients—requiring smaller step sizes and more steps to reach accurate solutions—our approach provides a useful idealized view of the process.
> > 3. **With CoT, this problem is indeed simple.** As shown in Figure 3, we observe empirically that with CoT, the Transformer converges to 100% accuracy within 60 SGD steps whereas training without CoT takes more than 19,500 steps in (n=30,k=4) parity problem. This is in line with our theoretical analysis, showing that CoT greatly simplifies the learning rate process for Transformer.
> >
> > \[1\] A Theory of Non-Linear Feature Learning with One Gradient Step in Two-Layer Neural Networks
> >
> > \[2\]  How one gradient step improves the representation.
> >
> > **Q3: Do we disable position embedding in our theoretical analysis and empirical experiments?**
> >
> > In our theoretical analysis, we composed the positional embedding and vocabulary embedding by assuming the embedding for each different boolean value at different positions to be different. This provides strictly more expressivity than positional embedding.  Our sample complexity lower bound (Theorem 1\) will still hold even with a position embedding so the sample complexity gap will still exist without CoT. Empirically, we have a trainable position embedding and still observe the large sample complexity gap with and without CoT.

---

> > > ### Comment · Reviewer_3yRE · 2024-11-26
> > >
> > > I appreciate the authors' response and revision, which helps to address my concerns. Despite limits of theoretical analysis, studying the convergence of zero-shot CoT is challenging interesting. I have increased my rating to 6. I suggest adding the answer to Q2 in the revision, since that convergence in three steps is counterintuitive and needs explanation when people first read it.

---

### Official Review · Reviewer_P7hs · 2024-11-04

**Soundness:** 3
**Presentation:** 4
**Contribution:** 4
**Rating:** 8
**Confidence:** 3

**Summary:**

This work aims to understand how Chain-of-Thought (CoT) improves sample complexity for transformers. In a setup of parity learning, it provides theoretical results of exponential sample complexity without CoT while the complexity of CoT is linear. Meanwhile, the training procedure with CoT induces sparse sequential patterns in attention, the rising of which is then empirically observed to be strongly correlated with perfect predicting in both settings of with and without CoT. The sparse attention patterns are also observed in finetuning on real-world data with CoT.

**Strengths:**

1. The motivation is good: understanding CoT is a significant direction to understand pre-training and finetuning LLMs, especially for reasoning tasks.
2. The theoretical setup is natural: parity learning is a popular setup for understanding neural networks and gradient methods, where some previous works were focusing on sampling complexity for staircase property in parity learning. Meanwhile, the sparse property of parity learning naturally brings the sparsity in learned attention.
3. Both theory and experiments clearly provide a exponential gap between the sample complexity with and without CoT.
4. The observation of sparsity in Qwen models on GSM8K is interesting, especially the math model finetuned with CoT.

**Weaknesses:**

1. In definition 2, the augmented sequence is with any permutation $S$. Is this permutation random or fixed during training? If it is random, it seems impossible to achieve perfect accuracy for next-token prediction in Theorem 3, and the monotone pattern in Figure 3(b) might also be questionable. If it is fixed, some clarification is helpful.
2. It would be better to include discussion regarding a recent related work [1].

[1] How Far Can Transformers Reason? The Globality Barrier and Inductive Scratchpad. Abbe et al. June 2024.

**Questions:**

Please see the above concerns.

---

> ### Author Response · Authors · 2024-11-24
>
> We thank the reviewers for recognizing the clear motivation of our work and the interesting results it provides.
>
> **W1: Is the permutation of indices in CoT sequence random or fixed during training?**
>
> The permutation $S$ is fixed during training. The reviewer's comment is entirely correct: If the permutation is not fixed, then it is impossible for any model to achieve perfect accuracy. We have provided further clarification in the revised version.
>
> **W2: Include discussion regarding a recent work.**
>
> We thank the reviewer for introducing this work and have added it to our related work section. In this work, the authors propose the concept of "globality degree", which shares a similar high-level intuition to the "sparsity dependence" we explore. The paper conjectures that a distribution is weakly learnable for transformers if and only if its globality is constant, and that scratchpads can enhance learning by breaking down the globality of the problem. These conjectures are mainly supported by extensive experiments, but lack theoretical proof. They also discuss the parity problem as a special example.
>
> While the overall message aligns with our own, there are key differences. First, we **theoretically** establish a separation between the sample complexity lower bound without CoT and the upper bound with CoT for the parity problem, whereas [1] offers only an empirical conjecture. Second, we demonstrate that CoT data induces **sparse attention** supported by theoretically analyzing the training dynamics as well as empirical experiments, an aspect not emphasized in [1].
>
> [1] How Far Can Transformers Reason? The Globality Barrier and Inductive Scratchpad. Abbe et al. June 2024.

---

### Official Review · Reviewer_vjNa · 2024-11-08

**Soundness:** 4
**Presentation:** 4
**Contribution:** 4
**Rating:** 8
**Confidence:** 4

**Summary:**

The paper studies how Transformers may or may not solve k-sparse parity problems. Without chain of thought, the authors provide a lower bound for one-pass algorithms with bounded memory, showing that exponential in k samples are needed, even though small transformers exist that solve the task. Then, the authors show positive results for learning with a CoT prompt using minibatch SGD, where order n samples are sufficient (where n is dimension, and up to log factors). The theory is complemented with extensive numerical experiments.

**Strengths:**

The results are interesting and novel, tackling the important question of training dynamics of transformers and the role of chain-of-thought prompting.

**Weaknesses:**

A couple of minor weaknesses (see also questions below):

* the lower bound is claimed to be about sample complexity, though it feels more like a computational lower bound. Perhaps this should be clarified.

* the initialization seems a bit non-standard, particularly W_{r,1:d}. Some more justification would be helpful.

**Questions:**

* could you comment on how the lower bound compares to other known computational lower bounds for parities, e.g. based on statistical queries, or in the spirit of Abbe-Sandon?

* could you clarify why the initialization on W_{r,1:d} in Assumption 1 is helpful, and whether it is needed? Is this what leads to "stronger correlation with the embedding ..." (L273) ? The phase 1 proof sketch would benefit from more clarity.

* note that O(n) learning of parities with fully-connected networks (thus with no sparse attention) is also possible when the staircase property holds (e.g. Abbe et al 2023, 2024). Is there a link between the CoT structure you consider and the staircase property?

* the proposed CoT prompt seems one of many possible choices, e.g. one could also have a length-n sequence that incrementally computes sparse parities up to each number. In a way, your way of encoding the problem almost requires that the user knows the secret set of locations, which seems pretty close to just giving the set as an input, which is a lot of information... Could you comment on any other possible choices, and why you chose this one?

minor typos: L247, "for any i" should be for any i, j and b vector?

---

> ### Author Response · Authors · 2024-11-23
>
> We appreciate the reviewer’s recognition of our work's novelty and significance. Below, we clarify and address your questions.
>
>  **W1 & Q1:** **It should be clarified that the lower bound is about sample complexity. Could you comment on how the established lower bound compares to other known lower bounds for parities?**
>
> As stated in Theorem 2, the lower bound we establish is indeed about **sample complexity**. Our result builds on **memory-sample** lower bounds from classical online learning communities [1] to show that $2^{\Omega (k)}$ samples are required to learn parity when the number of parameters is constrained to $o(nk)$, regardless of how much computation is put. There are indeed other lower bounds for learning parity in the literature, such as those under statistical query framework [2] and in Abbe and Sandon 2020 [3]. However, these lower bounds apply to setups distinct from ours. Below, we provide a detailed comparison of our results with these prior works.
>
> **Comparison with the statistical query model[2]:** It is well-known that $\Omega(n^k)$ queries are necessary in the SQ framework to learn parity [2]. In this framework, algorithms access an oracle that gives estimates of the expected value of some query function (for example, population loss gradient) over the data distribution. However, this framework **does not apply to the mini-batch SGD training** considered in our work, where gradients are averaged over small batches rather than the full data distribution. Our choice to focus on mini-batch SGD is motivated by two reasons: (1) Our primary goal is to analyze the **sample complexity** of learning parity with and without CoT, where population GD is not applicable. (2) mini-batch SGD is more aligned with real-world practice.
>
> **Comparison with Abbe and Sandon 2020 [3]**: [3] establishes two lower bounds for learning parity with one-pass SGD in different setups. The high-level idea is that under additional gradient noise or memory constraints, the output of a neural net trained on a parity function is **statistically indistinguishable** from the output of the neural net trained on junk data labels. However, these results rely on setups that diverge significantly from ours:
>
> - SGD with **additional randomness** [3, Theorem 6]: This result demonstrates that under **inverse-polynomial Gaussian noise**, polynomial-size neural networks fail to learn parity within polynomial steps of SGD. By contrast, our work assumes an accurate SGD oracle free of such gradient noise.
> - SGD with memory constraints: [3, Theorem 5 & Remark 6]: This result shows that when only $o(n / \log n)$ weights are updated per training step, polynomial-size networks fail to learn parity in polynomial steps of SGD. However, we consider standard SGD training where **$\Omega(n)$ parameters are updated simultaneously** in each training step.
>
> Furthermore, our lower bound applies to constant-pass training, while theirs only apply to one-pass SGD.
>
> To summarize, our lower bound focuses on constant-pass mini-batch SGD training without additional noise when the number of parameters is limited ($o(nk)$), but all parameters are updated simultaneously in each training step, which is different from the above setups. We have revised our related work section to include a discussion of these differences
>
> [1] Xin Lyu, Avishay Tal, Hongxun Wu, and Junzhao Yang. Tight time-space lower bounds for constantpass learning, 2023.
> [2]Kearns. Efficient noise-tolerant learning from statistical queries, 1998.
> [3]Emmanuel Abbe and Colin Sandon. Poly-time universality and limitations of deep learning, 2020.

---

> ### Author Response · Authors · 2024-11-24
>
> **W2 & Q2:  Justification for Initialization; Why the initialization on $W_{r,1:d}$ in is helpful?**
>
> **Why the initialization on $W_{r,1:d}$ in is helpful?** The initialization we used on the FFN weights $W_{r,1:d}$ is $W_{r,1:d}=\sum_{i=n+1}^{n+k}\sum_{b=0}^1 v_{r,i,b}e_{i,b}$, where $v_{r,i,b}\sim U \\{ -\epsilon,\epsilon \\}$. This initialization ensures that the activation pattern of the FFN (i.e. whether the $r$-th neuron is activated by the ReLU activation) is solely determined by the position $i$, the boolean value $\mathbf b[i]$ and the randomness of $v_{r,i,\mathbf b[i]}$ since $\langle e_{i,\mathbf b[i]},W_{r,1:d}\rangle \approx v_{r,i,b}$ with high probability due to near-orthogonal embedding. As a result, for a fixed $(i,\mathbf b[i])$, the FFN can be viewed as a linear function, which consists of the neurons with $v_{r,i,\mathbf b[i]}>0$. At phase 1, this linear function will develop a stronger correlation with embedding $e_{(S[i],\mathbf b[j])}$.
>
> The explanation is mainly provided in the proof sketch of Phase 1. We've provided further explanation in the revised version.
>
> **Justification for the initialization:** Our analysis relies on the embeddings $e_{i,b}$ being near-orthogonal to each other. If we assume the embeddings $e_{i,b}$ are exactly orthogonal, such as one-hot vectors, then the initialization of $W_{r,1:d}$ involves randomly sampling each coordinate from $\{-\epsilon, \epsilon\}$, which is a standard approach. To accommodate the more general case where $e_{i,b}$ are not strictly orthogonal, we rotate to a non-orthogonal basis defined by $\{e_{i,b}\}$, which simplifies our analysis.

---

> > ### Author Response · Authors · 2024-11-24
> >
> > **Q3 Discussion on the staircase property [3,4]**
> >
> > We first recall the concepts of merged-staircase property and leap complexity introduced in [3] and [4] respectively. A Boolean function satisfies the merged-staircase property if its non-zero Fourier coefficients $ \\{ S_i \\}_{i \in [1,m]}$ can be ordered such that for each $i$ , the set difference $|S_i \setminus \cup\_{i' < i} S\_{i'}| \leq 1$. Furthermore, the leap complexity of this boolean function is defined as the smallest $\ell $ such that after reordering,  $|S_i \setminus \cup\_{i' < i} S\_{i'}| \leq \ell$ for each $i$. In [3], it is proved that the merged-case property is a necessary and nearly sufficient condition for efficient learning in 2-layer neural networks within the mean-field regime. In [4], it is conjectured that an FFN can efficiently learn a boolean function if and only if the leap complexity is a constant.
> >
> > The reviewer commented that "O(n) learning of parities with fully-connected networks is also possible when the staircase property holds". However, we would like to point out that the **merged-staircase property does not hold for parity function (without CoT)**. Thus learning parity requires exponential steps, as proved in [3] for 2-layer neural networks within the mean-field regime, and conjectured in [4] for more general setups.
> >
> > When incorporating CoT, learning parity becomes an **autoregressive** task, which lies outside the scope of [3,4] as they consider **one-turn prediction** with **FFN**. But conceptually, CoT simplifies the problem by reducing the leap complexity for predicting the next token from $k$ to a constant (specifically, $2$) (Although the **merged stair-case property still does not hold**). Extending concepts like the staircase property and leap complexity to autoregressive tasks is an exciting direction for future research.
> >
> > **Q4 Why choose the proposed CoT prompt?**
> >
> > **CoT sequences give a lot of information?** We would like to emphasize that, **information-theoretically**, $O(n)$ samples (without CoT) already contain sufficient information to learn parity, as Gaussian elimination can be applied. The challenge lies in efficiently extracting this information (due to the established lower bound in Theorem 2). CoT sequences do provide some guidance that makes learning the parity problem more efficient, which reflects the essence of training with CoT data: CoT decomposes a difficult task into simpler intermediate steps and provides clear instructions on how to solve them. Similar CoT formats—explicitly emitting intermediate computation steps—are commonly used in CoT-based approaches for arithmetic problems: For instance, [5] demonstrates the approach for long addition problems, [6] shows that binary tree-structured decompositions facilitate learning parity in RNNs, and [7] adopts a CoT format identical to ours.
> >
> > **Why did we choose this particular CoT format?**
> >
> > We agree that other CoT formats could also be effective. For example, the length-$n$ CoT suggested by the reviewer is a plausible alternative, as empirical experiment shows that this CoT format can also be efficiently learned. Binary tree-structured decompositions, as in [6], could be another option. These formats also exhibit sparse dependencies and theoretical analysis may be possible as well. We chose this CoT format primarily for its simplicity for theoretical analysis since each bit in this CoT is derived through a fixed simple rule (i.e. it is the XOR of the previous bit and a secret bit).
> >
> > [5] Nye et al. Show your work: Scratchpads for intermediate computation with language models. 2021.
> >
> > [6] Wie et al. Sub-task decomposition enables learning in sequence to sequence tasks.  2023.
> >
> > [7] Abbe et al. How Far Can Transformers Reason? The Globality Barrier and Inductive Scratchpad. 2024.
> >
> > Regarding the additional experiment on length-$n$ CoT, we considered a setup analogous to Figure 3(a). Specifically, we trained a 1-layer, 1-head transformer on the parity problem with $n=100$ while varying $k$ and shows that this CoT have similar sample complexity as ours empirically. The detailed sample complexity is as follows:
> >
> > | k                 | 20                 | 30                 | 40                 | 50            | 60                 | 70                 | 80                 | 90                 |
> > | ----------------- | ------------------ | ------------------ | ------------------ | ------------- | ------------------ | ------------------ | ------------------ | ------------------ |
> > | sample complexity | $1.42\times10^{5}$ | $1.79\times10^{5}$ | $1.57\times10^{5}$ | $1.60*10^{5}$ | $1.55\times10^{5}$ | $1.61\times10^{5}$ | $1.39\times10^{5}$ | $1.59\times10^{5}$ |

---

### Meta-Review · Area_Chair_87ma · 2024-12-23

**Metareview:**

This paper investigates how Chain-of-Thought (CoT) enhances the sample efficiency of Transformer models in reasoning tasks, focusing on the parity learning problem. The authors theoretically demonstrate an exponential gap in sample complexity between learning with and without CoT for the parity task. They show that CoT induces sparse sequential dependencies and leads to sparse, interpretable attention patterns. These theoretical findings are validated through experiments on both synthetic and real-world datasets.

The paper's primary strength lies in its novel theoretical analysis of Transformer learning dynamics, particularly challenging due to the architecture's non-linear nature. The rigorous proof of the exponential sample complexity gap between CoT and non-CoT learning for parity tasks is a significant contribution. The work also provides valuable insights into the connection between CoT and the induction of sparse attention patterns, supported by empirical validation on both synthetic and real-world datasets.

While the theoretical analysis is robust, some reviewers initially found parts of the proofs difficult to follow, though this was largely addressed in the author response. The theoretical setup uses some simplifying assumptions, such as a fixed secret for the parity task, which may limit generalizability. Additionally, while supportive, the real-world experimental results could be stronger to fully validate the theoretical claims.

Despite these minor weaknesses, the paper makes a significant theoretical contribution to understanding the mechanics of CoT in Transformer models. It provides a rigorous foundation for explaining the empirically observed benefits of CoT in LLMs, combining theoretical analysis with empirical validation. The authors have been responsive to reviewer concerns, improving the clarity and presentation of their work. These factors strongly support the decision to accept the paper.

**Additional Comments On Reviewer Discussion:**

During the rebuttal period, reviewers raised several key points which the authors addressed comprehensively. Concerns about the clarity of proofs were addressed by restructuring and adding more details, improving readability. The use of a fixed secret for the parity task was justified as a minimal example for studying CoT's impact on sample complexity. Questions about the theoretical three-step convergence were explained as an idealized view common in theoretical studies, with empirical results showing significantly faster convergence with CoT compared to without.

Reviewers questioned the significance of the attention entropy differences in real-world experiments, to which the authors provided additional context highlighting consistent trends across layers and models. The role of the initialization scheme was clarified, explaining its purpose in guaranteeing certain properties of the MLP activation patterns. Lastly, the authors addressed a suggestion to discuss a recent related paper, adding it to their related work section and clarifying the distinctions of their approach.

Overall, the authors' thorough responses and willingness to improve the paper addressed the main concerns raised by the reviewers, particularly in improving the clarity of their proofs and justifying their methodological choices.

---

### Decision · Program_Chairs · 2025-01-22

Accept (Poster)